# Aerosol indirect effects on the temperature-precipitation scaling

Nicolas Da Silva[1], Sylvain Mailler[1,2], and Philippe Drobinski[1]

[1]LMD/IPSL, École polytechnique, Université Paris Saclay, ENS, PSL Research University, Sorbonne Universités, UPMC Univ Paris 06, CNRS, Palaiseau, France
[2]ENPC, Champs-sur-Marne, France

**Correspondence:** Nicolas Da Silva, Centre for Ocean and Atmospheric Sciences, School of Environmental Sciences, University of East Anglia, Norwich, Norfolk, NR4 7TJ, United Kingdom, n.da-silva@uea.ac.uk

**Abstract.**

Aerosols may impact precipitation in a complex way involving their direct and indirect effects. In a previous numerical study, the overall microphysical effect of aerosols was to weaken precipitation through reduced precipitable water and convective instability. The present study aims at quantifying the relative importance of these two processes in the reduction of summer
precipitation using the temperature-precipitation scaling. Based on a numerical sensitivity experiment conducted over central Europe aiming to isolate indirect effects, all others effects being equal, the results show that the scaling of hourly convective precipitation with temperature follows the Clausius-Clapeyron (CC) relationship whereas the decrease of convective precipitation does not scale with the CC law since it is mostly attributable to increased stability with increased aerosols concentrations rather than to decreased precipitable water content. This effect is larger at low surface temperatures for which clouds are sta-
tistically more frequent and optically thicker. At these temperatures, the increase of stability is mostly linked to the stronger reduction of temperature in the lower troposphere compared to the upper troposphere which results in lower lapse rates.

## 1   Introduction

The temperature-precipitation relationship has often been studied because it has been hypothesised to give an insight of the
change of precipitation in a warming climate. In this context, one may distinguish extreme precipitation studies from mean precipitation studies. The Clausius-Clapeyron (CC) law relates changes in temperature to changes in water vapor content assuming constant relative humidity:

$$\frac{\partial e_s}{\partial T} = \frac{L_v e_s}{R_v T^2} \tag{1}$$

where $e_s$ is the water vapor saturation pressure, $T$ is the temperature, $L_v$ is the latent heat of vaporization and $R_v$ is the gas
constant for air. It has been suggested that precipitation extremes correspond to events where the whole column of moisture is precipitated out and are therefore expected to scale with the CC law (Pall et al., 2007; Muller, 2013). However many

departures from the CC-scaling have been observed. Literature has described a peaklike shape for the temperature-precipitation extremes relationship with CC-scaling for the cold season and negative scaling for the warm season (Drobinski et al., 2016). Sub-CC scaling for warm temperatures can be explained by either the decrease of relative humidity (Hardwick et al., 2010;
Panthou et al., 2014), the decrease of precipitation duration (Utsumi et al., 2011; Singleton and Toumi, 2013; Panthou et al., 2014), the decrease of precipitation efficiency or changes in dynamics (Drobinski et al., 2016). Conversely, Lenderink and van Meijgaard (2008) has found an increase of precipitation extremes (their 99.9th and 99th percentiles) beyond the CC-scaling for temperatures between 12°C and 23°C at De Bilt in Netherlands. It has been argued that this "super-CC" scaling is due to the transition between stratiform and convective precipitation (Haerter and Berg, 2009; Berg and Haerter, 2013; Molnar et al.,
2015) and enhanced dynamics in convective clouds at higher temperatures (Lenderink et al., 2017). Although less documented than extremes, a "hook shape" of the temperature-precipitation relationship, that is a positive slope at low temperatures and a negative slope at high temperatures, is also suggested for mean precipitation (Zhao and Khalil, 1993; Madden and Williams, 1978; Crhová and Holtanová, 2017; Rodrigo, 2018) as well as differences between land and sea areas (Adler et al., 2008; Trenberth and Shea, 2005). The CC scaling is less expected for global mean precipitation which are more constrained by an
energetic budget than extreme precipitation (Allen and Ingram, 2002; Held and Soden, 2006; Muller et al., 2011; Muller, 2013). On the local scales, the energy budget includes a term accounting for the transport of dry static energy which may suppress the constraint in many regions, as shown by Muller and O'Gorman (2011). The study of Hardwick et al. (2010) suggests that Australia is part of the regions were the constraint still holds. Indeed, they have systematically found lower slopes for median precipitation than for extreme precipitation in their 4 selected in-situ measurement stations in Australia.

The fact that the CC law is not always adequate for describing the temperature-precipitation relationship in a given climate does not mean that if one would perturb the climate, the change in precipitation would not follow a CC-scaling. Indeed, using Regional Climate Models (RCM) in the Mediterranean region and within the frame of the HyMeX program (Drobinski et al., 2014), Drobinski et al. (2018) found a CC-scaling between past and future climate while observing hook shapes for both past and future climate temperature-precipitation relationships. It has often been shown that extreme precipitation would increase
at a rate similar to the CC law whereas mean precipitation would increase at a lower rate in a warming climate (Allen and Ingram, 2002; Boer, 1993; Trenberth, 1998; Held and Soden, 2006).

Apart from the greenhouse gases forcing, the forcing of aerosols is another feature that can modify climate and therefore temperature-precipitation relationship. Aerosols affect climate through their direct and semi-direct effects as well as through their effects on cloud microphysics (indirect effects). While their direct effect is rather well understood, many uncertainties
remain for the indirect effects. Stevens and Feingold (2009) described aerosol cloud interactions as a buffered system in which many processes seem to partly compensate each other. Among these effects, the Twomey (1977) effect, also called "first indirect effect", is an increase of the Cloud Optical Depth (COD) through reduced cloud droplet radius for constant liquid water content with increased aerosol concentrations. Aerosols indirect effects may also increase cloud lifetime (Albrecht, 1989) but as of today no consensus exists on the reality of this effect (Small et al., 2009; Seifert et al., 2015; Malavelle et al., 2017),
and its representation in climate models is highly dependent on the model's microphysical formulation (Zhou and Penner, 2017). Many hypotheses have been suggested for the effect of aerosols in convective clouds such as the convective invigoration

(Khain et al., 2004, 2005; Rosenfeld et al., 2008; Lebo and Seinfeld, 2011; Fan et al., 2013) which would be the consequence of an increased release of latent heat due to ice formation associated with a decrease of warm rain formation with increased aerosol loads. Miltenberger et al. (2018) have however observed an invigoration of deep convective clouds below the freezing level, indicating that the previous theory may not hold under certain conditions. Instead, they proposed that the observed enhancement of precipitation once convection is organized, is caused by reduced dry air entrainment in the convective core of polluted clouds. Aerosols were also hypothesised to affect precipitation through changes in cloud evaporation (Xue and Feingold, 2006; Dagan et al., 2015; Liu et al., 2019) or through changes in entrainment rates (Dagan et al., 2015; Miltenberger et al., 2018).

A common feature of both direct and indirect effects of aerosols is a global decrease of precipitation through a decrease of evaporation from the surface due to the reduction of shortwave downwelling fluxes at the surface (Ramanathan et al., 2001; Lelieveld et al., 2002; Bollasina et al., 2011; Salzmann et al., 2014). Many modeling case studies may underestimate this longer-term feedback performing short-time simulations or simulations in a small domain (Khain et al., 2004, 2005; Khain and Lynn, 2009; Lebo and Seinfeld, 2011; Lebo and Morrison, 2014; Lebo, 2014; Miltenberger et al., 2018). Conversely, long-term simulations are often performed at a coarse resolution using convective parameterizations that do not implement the microphysical effect of aerosols on convective clouds (Bollasina et al., 2011; Salzmann et al., 2014). Few studies take into account both of the microphysical effects of aerosols on convection and the aerosol long-term radiative feedback by realising high resolution simulations over few months (Morrison and Grabowski, 2011; Fan et al., 2013), while not avoiding uncertainties related to their representation by RCM in these intermediate configurations. The study of Da Silva et al. (2018) over the Euro-Mediterranean area is one of them in which the decrease of surface precipitation was related to the radiative path previously described (see Fig. 1). The authors have shown that the consecutive surface cooling not only reduces the water content but also stabilises the atmosphere as suggested by Fan et al. (2013); Morrison and Grabowski (2011); Stjern et al. (2017), and hence acts in reducing precipitation with increased aerosol concentrations. A third path is possible as a combination of these two paths since the reduction of water vapor mixing ratio at the surface would also contribute to increase the stability of the atmosphere through less latent heat released with increased aerosol concentrations. To our knowledge, an evaluation of the relative contribution of these paths to precipitation reduction due to aerosol indirect effects has not been proposed yet. This study aims at determining these contributions and therefore can be seen as a natural follow-up of Da Silva et al. (2018). For that purpose, we use the temperature-precipitation relationship which appears to be a natural framework since both effects are a consequence of the decrease of surface temperature.

Section 2 details the configuration of the WRF model used, the simulations, and the method that have been performed for this sensitivity analysis. Section 3 analyses the temperature-precipitation scaling and quantifies each contribution to the reduction of central Europe summertime precipitation under the effect of a massive concentration of cloud condensation nuclei. Section 4 concludes the study.

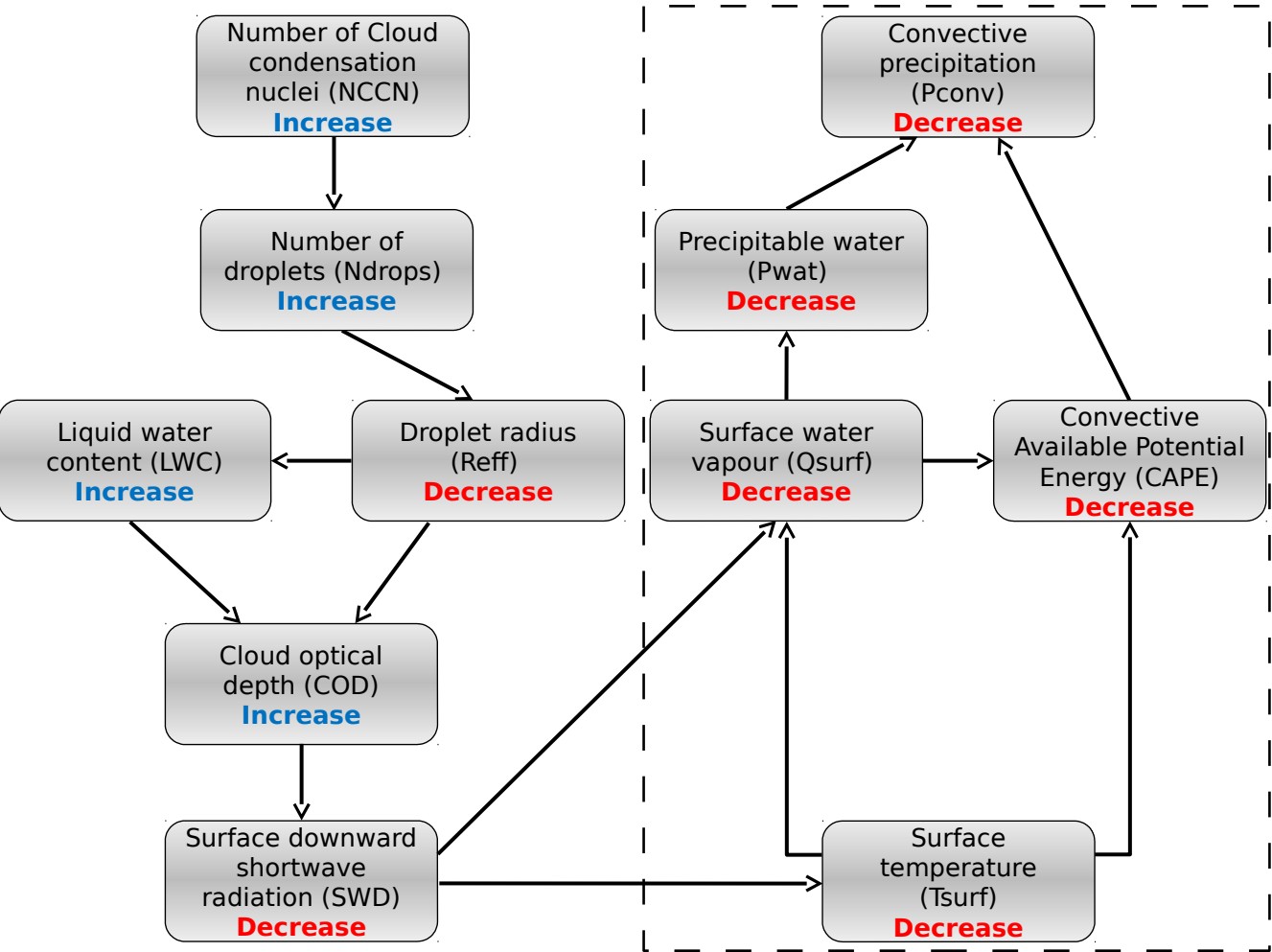

**Figure 1.** Schematic summary of the aerosol causal sequence for the indirect effects of aerosols on convective precipitation (adapted from Da Silva et al., 2018). The dotted rectangle indicates the part of the scheme which is detailed in the present study.

## 2 Methods

### 2.1 Model configuration

The version 3.7.1 of the Weather Research and Forecasting Model (WRF, Skamarock et al., 2008) is used in this study. The model was run with a 50 km (LR), a 16.6 km (MR), and a 3.3 km (HR) horizontal resolution on a domain displayed in Fig. 2. It is forced by the Global Forecast System (GFS) model (National Centers for Environmental Prediction National Weather Service, 2000) as initial and boundary conditions. Temperature, humidity, geopotential and velocity components are nudged towards

GFS analysis data with a Newtonian-type method using a relaxation coefficient of $5 \times 10^{-5}\,\mathrm{s}^{-1}$ as recommended by, e.g., Salameh et al. (2010); Omrani et al. (2013, 2015).

    The microphysical scheme used is the Thompson and Eidhammer (2014) scheme which explicitly calculates the number concentrations of aerosols. The latter are represented in a simplified way according to their capacity to nucleate cloud water ("water friendly", WFA) or ice water ("ice friendly", IFA). Aerosol number concentrations are initialized and forced at domain

boundaries by a climatology based on Goddard Chemistry Aerosol Radiation and Transport (GOCART) model (Ginoux et al., 2001) simulations. While no surface emissions are applied to IFA, surface emission fluxes are applied to WFA in order to approximately equilibrate the loss of WFA due to scavenging and nucleation. The radiation scheme is RRTMG (Rapid Radiative Transfer Model for General circulation models, Iacono et al., 2008) and uses the cloud water droplets, ice and snow effective radii of the Thompson and Eidhammer (2014) microphysical scheme to resolve the radiative transfer equations. Another cli-

matology of aerosols from Tegen et al. (1997) is used in this radiative scheme and therefore is not affected by any changes in the microphysical aerosol climatology, which enables us to perform sensitivity experiments of the indirect effects of aerosols with fixed aerosol direct effect. The Kain (2004) scheme is used to parameterize convection. The microphysical effects of aerosols are not taken into account explicitly in this parameterization although they can affect convection indirectly through modifications in the temperature or moisture profiles.

This configuration is the same as in Da Silva et al. (2018) to which the reader is referred for additional detail.

## 2.2   Simulation experiments

    The model was run to make two extreme simulations in terms of WFA and IFA microphysical concentrations. Both simulations start on April $1^{\mathrm{st}}$, 2013 (after one month of spin-up) and end on September 17, 2013. A very high aerosol emission level ($1.75 \times 10^7$ kg s$^{-1}$ for the whole domain) is applied in the first simulation, referred as MAX or polluted simulation and a very low

aerosol emission level ($1.75 \times 10^{-4}$ kg s$^{-1}$ for the whole domain) is applied for the other simulation, referred as MIN or pristine simulation. Although these emission rates are extreme, maximal and minimal value permitted by the microphysics scheme reduce the range of variation of the number of WFA (NWFA) between $\sim 10\,\mathrm{cm}^{-3}$ and $\sim 10,000\,\mathrm{cm}^{-3}$ and of the number of IFA (NIFA) between $0.005\,\mathrm{cm}^{-3}$ and $10,000\,\mathrm{cm}^{-3}$. Therefore these latter extreme emission rates ensure that both NIFA and NWFA in the MIN (resp. MAX) simulation remain close to their minimal (resp. maximal) permitted values, which corresponds

to a $2 \times 10^6$ factor for NIFA and a $10^3$ factor for NWFA between the MAX and the MIN simulations. Such high differences of aerosol concentrations between the two simulations ensure that aerosol indirect effects are strong enough to emerge from the potential noise between the MAX and the MIN simulations. On the flip side, extreme values of aerosol concentrations reach the bounds of permitted values in the microphysical scheme, suggesting that for these ranges of concentrations, the uncertainties associated to the parameterizations of microphysical processes may be more pronounced.

Another set of MIN and MAX simulations has been performed at a resolution where convection is resolved (3.3 km) and on a smaller domain (HR domain) as seen in Fig. 2. An intermediate set of simulations was used to perform one-way nesting between the LR and the HR simulations, ensuring that the LR simulations force the HR simulations at their boundaries. These intermediate simulations were done at 16.6 km of resolution in an intermediate domain (MR, see Fig.2) and with the same

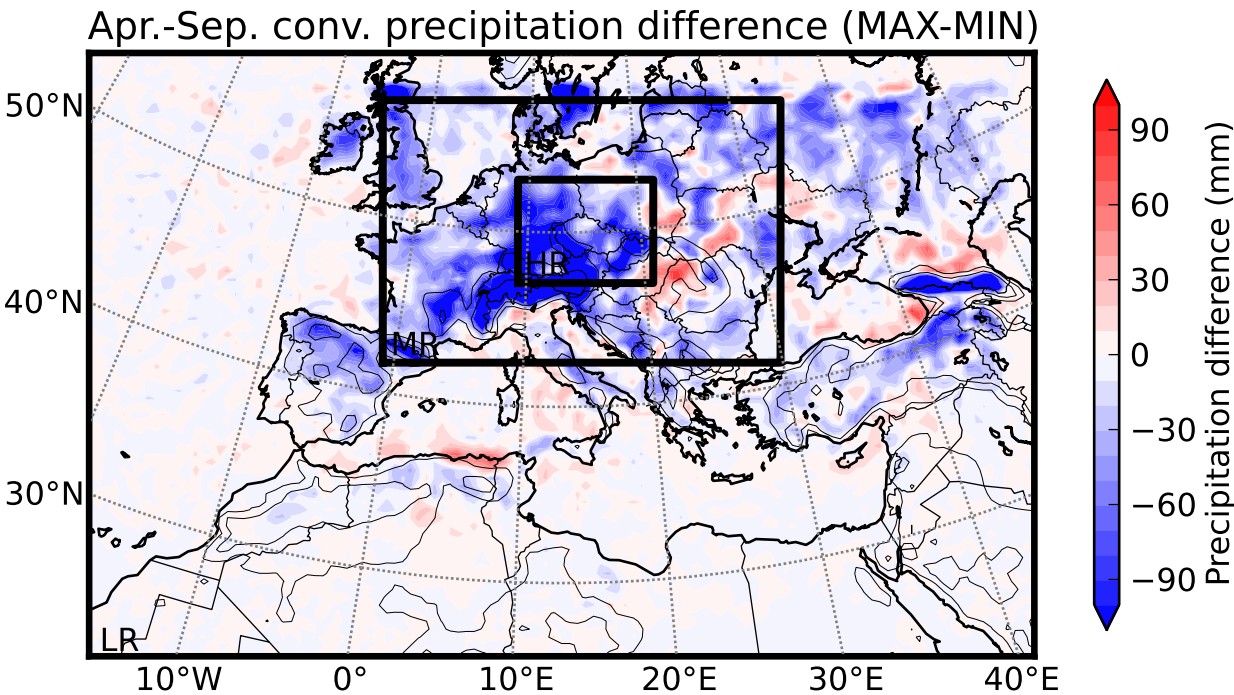

**Figure 2.** Differences of convective precipitation between the MAX and the MIN simulations. The whole map is the LR simulation domain, the medium black box is the intermediate domain MR, and the small box is the HR domain.

configuration as the LR simulations. In these conditions, each grid cell of the LR domain corresponds to exactly $15 \times 15$ grid cells of the HR domain. The HR simulations have been performed without activating any convection scheme, since horizontal resolution (3.3 km) is sufficient to resolve convection processes, which is the only difference in model configuration between the LR simulations and the HR simulations. While the microphysical effects of aerosol on convective clouds were not taken into account due to the use of a convection scheme insensitive to aerosol concentrations, the whole set of indirect effects are represented in the HR simulations, including small scale and large scale processes.

## 2.3 Temperature-precipitation bin method

The simulation domain covers the Euro-Mediterranean region as displayed in Fig. 2. This figure also shows the difference of accumulated convective precipitation over the period of study between the MAX and the MIN simulations. It shows that most of the negative signal is concentrated over land regions where precipitation are more intense in this period of the year (Da Silva et al., 2018). The following analysis of convective precipitation reduction in the MAX simulation is conducted over

the HR domain. Indeed, location of the HR domain was chosen because of the high negative values of convective precipitation differences between the MAX and the MIN simulations in this area and because it is far away from oceanic areas where flux imbalance with the non-coupled oceanic surface may hinder interpretation as discussed in Da Silva et al. (2018). Because of the short duration of our simulations, temperature at first vertical grid level (centered around 28 m above the ground, hereafter referred to as surface) and convective precipitation hourly time series were collected for all grid points of the WRF model that

were inside the HR domain and then concatenated. To avoid snow precipitation we selected only the events with daily mean temperatures warmer than 5°C.

    The method used to scale precipitation with temperature is similar to the one used by Hardwick et al. (2010). Temperature has a diurnal variation and may be impacted by precipitation events. Since for each precipitation event we want the corresponding temperature that represents the air mass, the daily averaged temperature is used. We select hours with strictly positive precip-

itation amount in both the MIN and MAX time series and place the pairs of daily mean temperatures and hourly precipitation into 8 bins of 5896 samples according to the daily temperatures. In each bin the 50th percentile of daily mean temperature, the 50th percentile of precipitation and the 95th percentile of precipitation are used for our analysis.

    We focus on the contributions of precipitation efficiency, surface water vapor mixing ratio, and maximum vertical wind speed to the difference of convective precipitation scaling with temperature between the MAX and the MIN simulations. We define

precipitation efficiency as the ratio between the total mass of condensate of a column and the effective rate of precipitation that reach the surface. For the LR simulations, it is calculated using hourly output variables of WRF, and following the parameterization of Kain (2004) implemented in the model in which precipitation efficiency is a decreasing function of cloud base height and vertical wind shear. Because model output frequency is lower than the typical convective characteristic time, we expect large uncertainties. Precipitation efficiency is not explicitly calculated in the HR simulations, we therefore estimated it using

the ratio of precipitation divided by the product of maximum vertical wind speed and surface water mixing ratio. For the LR simulations, the maximum vertical wind speed is calculated using the square root of surface based Convective Available Potential Energy (CAPE) which is more representative of convective vertical motions than the resolved vertical velocity. These three variables are computed one hour before the convective precipitation occurrence to better represent the air inside the updraft of the convective cell rather than the air inside its downdraft.

The contribution of each variable to the change of precipitation between the MAX and MIN simulations is computed for both median and extreme precipitation events which are defined as following. Median events are all events where precipitation is between the 40th and the 60th percentile in at least one of the simulations (MIN or MAX). Extreme events are all events beyond the 90th percentile in at least one of the simulations (MIN or MAX). Median and extreme events are sorted as a function of the corresponding daily mean temperature and placed in 8 bins with the same number of events per bin. For median or

extreme precipitation, the median of daily mean temperature is paired with each of the 4 variables (precipitation, precipitation efficiency, surface water vapor mixing ratio and maximum vertical wind speed along the atmospheric column) in the MIN and the MAX simulations.

## 3 Results

### 3.1 Sensitivity of temperature-precipitation scaling to change in aerosol loads

Figure 3 displays the 50[th] (a, c) and 95[th] (b, d) percentiles of hourly convective (a, b) and total (c, d) precipitation as a function of daily mean temperature at the surface for both the LR MIN (magenta) and the LR MAX (blue) simulations. Median total precipitation displays a negative scaling with surface temperature for both LR and HR simulations (Fig. 4a). Since the temperature range is spread over 2 seasons, it is likely that changes in large scale forcings between spring and summer events may explain the decrease of median precipitation with surface temperature. Sub-CC scaling for median total precipitation are

consistent with the study of Hardwick et al. (2010) in Australia. On the other hand, median convective precipitation follow a nearly CC-scaling in our LR simulations indicating that, unlike median total precipitation events, convective precipitation events seem to be mostly affected by changes in surface temperatures rather than changes in large scale dynamics.

Regarding convective precipitation extremes, a nearly CC-scaling appears in the LR simulation. Using in-situ measurements in Switzerland, Molnar et al. (2015) found a scaling of $8.9\%.^{\circ}C^{-1}$ of hourly convective precipitation as a function of daily mean

temperature. Lower but similar slopes are obtained in our study with a value of $6.1\%.^{\circ}C^{-1}$ for the LR MIN simulation and a value of $8.6\%.^{\circ}C^{-1}$ in the LR MAX simulation. Berg and Haerter (2013) and Loriaux et al. (2013) showed that the scaling between total extreme precipitation and daily mean temperature could be super-CC because of the distribution of convective and stratiform precipitation with respect to daily mean temperature. Convective precipitation are generally more intense and occur at higher temperatures. Supposing that both convective and stratiform precipitation follow a CC-scaling, they argued

that total precipitation will display a super-CC scaling for temperatures corresponding to the transition between stratiform and convective precipitation. Such an effect does not appear in our study since we can observe a slight sub-CC scaling for total extreme precipitation. The scaling of total extreme precipitation is therefore different from the hook shape found in the Drobinski et al. (2018) study in the Mediterranean area. As expected (Li et al., 2011), precipitation extremes are increased in the HR simulations with respect to the LR simulations. However the slopes of the HR simulations are rather similar to the

slopes of total precipitation in the LR simulations.

Differences between the MAX and the MIN simulations are similar for both extremes and medians in HR and LR simulations. We find that convective precipitation are reduced in the MAX simulation but only at low temperatures. This temperature dependency slightly changes the scaling between the MAX and the MIN simulations, with higher slopes in the MAX simulation (around $8.5\%.^{\circ}C^{-1}$ in LR) compared to the MIN simulation (around $6.2\%.^{\circ}C^{-1}$ in LR). The fact that indirect effects of aerosols

are weaker at high temperatures is probably due to the lower occurrence of clouds in these conditions. Figure 5 shows mean COD calculated as in Da Silva et al. (2018), as a function of daily mean temperature for both the MIN and MAX simulations for low and high resolutions. It shows a decrease of COD with temperatures in all of the simulations. When averaging over hours with strictly positive COD, we found that the optical thickness of clouds is relatively constant over the temperature range (not shown), confirming that the decrease of COD with temperature is mostly due to a decrease of the occurrence of

clouds with temperature. This tendency maximizes the indirect effects of aerosols at low temperatures and minimizes them at high temperatures. In their study of the impact of the microphysical scheme on the scaling of precipitation extremes with

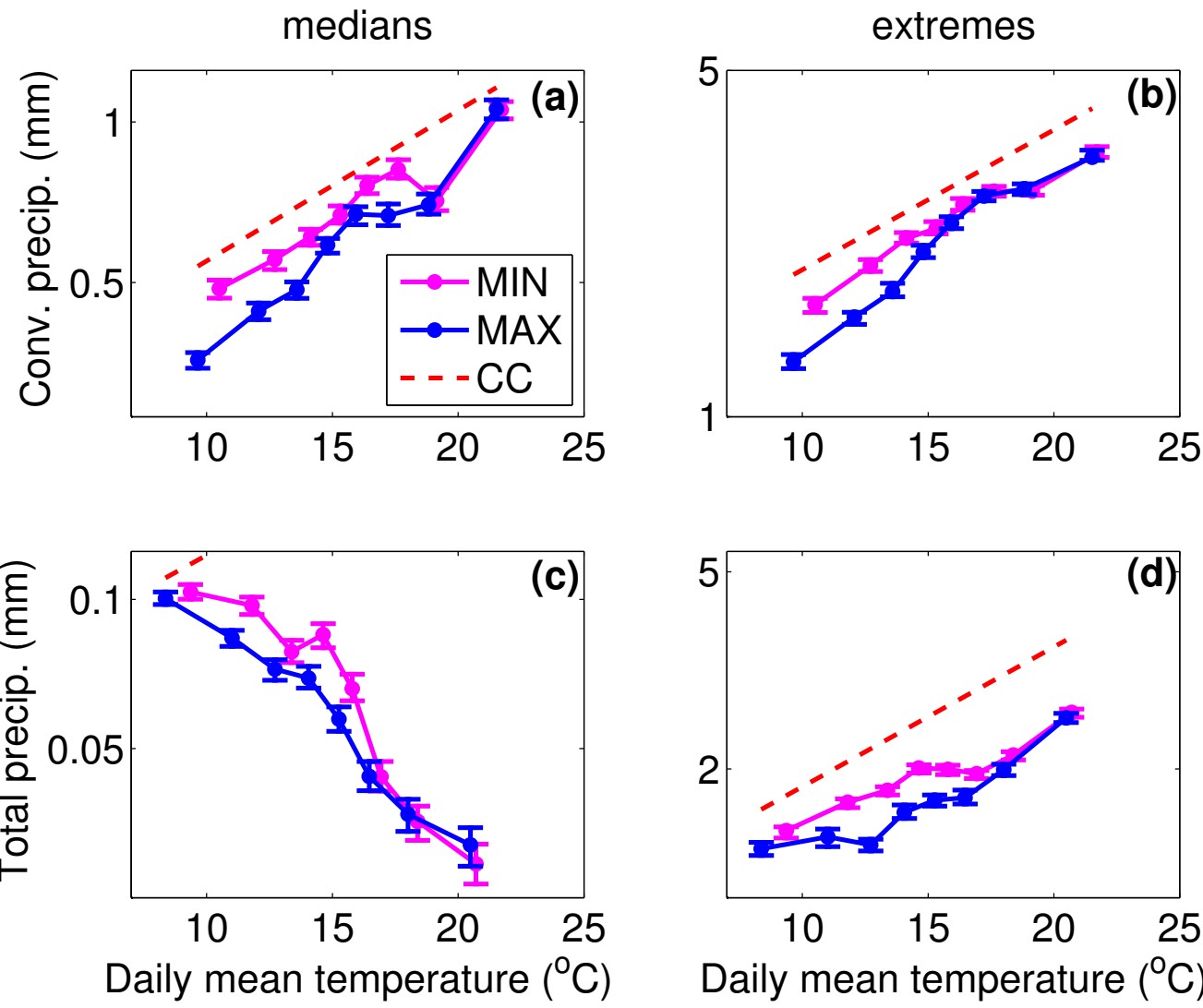

**Figure 3.** Hourly convective (a, b) and total (c, d) precipitation as a function of daily mean temperature at the surface for median (a, c) and extreme (95[th] percentile, b, d) precipitation and for both the LR MIN (magenta) and LR MAX (blue) simulations. The dashed red line indicates the CC-slope calculated using the August-Magnus-Roche approximation for saturated vapor pressure (Alduchov and Eskridge, 1996). Errorbars represent the 95 % confidence interval of the precipitation percentiles.

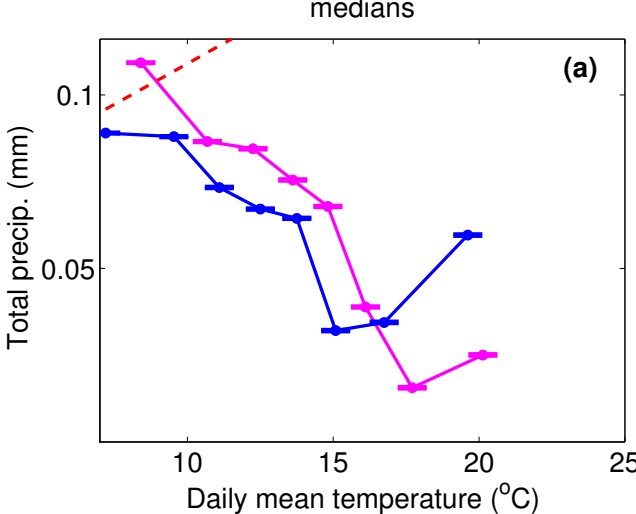
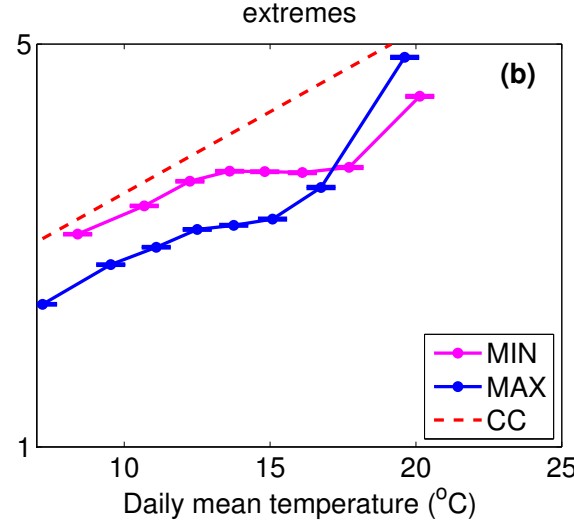

**Figure 4.** Hourly total precipitation as a function of daily mean temperature at the surface for median (a) and extreme (95[th] percentile, b) precipitation and for both the HR MIN (magenta) and HR MAX (blue) simulations. The dashed red line indicates the CC-slope calculated using the August-Magnus-Roche approximation for saturated vapor pressure. Errorbars represent the 95 % confidence interval of the precipitation percentiles.

temperature, Singh and O'Gorman (2014) have also shown that the main effect occurs at low temperatures. They attributed the change of slope at low temperatures to a difference in the parameterization of hydrometeor fall speed. They argued that slower hydrometeor fall speed decreases precipitation efficiency through enhanced evaporation and detrainment and through reduced

precipitation rate. In our case, convective precipitation are diagnosed with the same convective scheme in the LR MAX and LR MIN simulations, which neither takes into account aerosol concentrations nor rain fall speed. Such microphysical effect is therefore impossible for the LR simulations. In the HR simulations however, convective precipitation are diagnosed by the microphysical scheme which does represent the terminal fall speed of hydrometeors that may be different depending on the aerosol concentrations, as found in other studies (Koren et al., 2015; Dagan et al., 2018). This effect is one among other that

possibly acts in changing the precipitation efficiency between the HR MAX and the HR MIN simulations. The next section is dedicated to evaluate how strong are these changes in precipitation efficiency relatively to the background radiative effect between the MIN and the MAX simulations.

### 3.2   Process analysis

To analyse the reduction of convective precipitation at low temperatures we consider that precipitation can be approximately

described by the following equation (Drobinski et al., 2016; Da Silva, 2018):

$$Pr \propto \epsilon \times Q \times W \qquad\qquad (2)$$

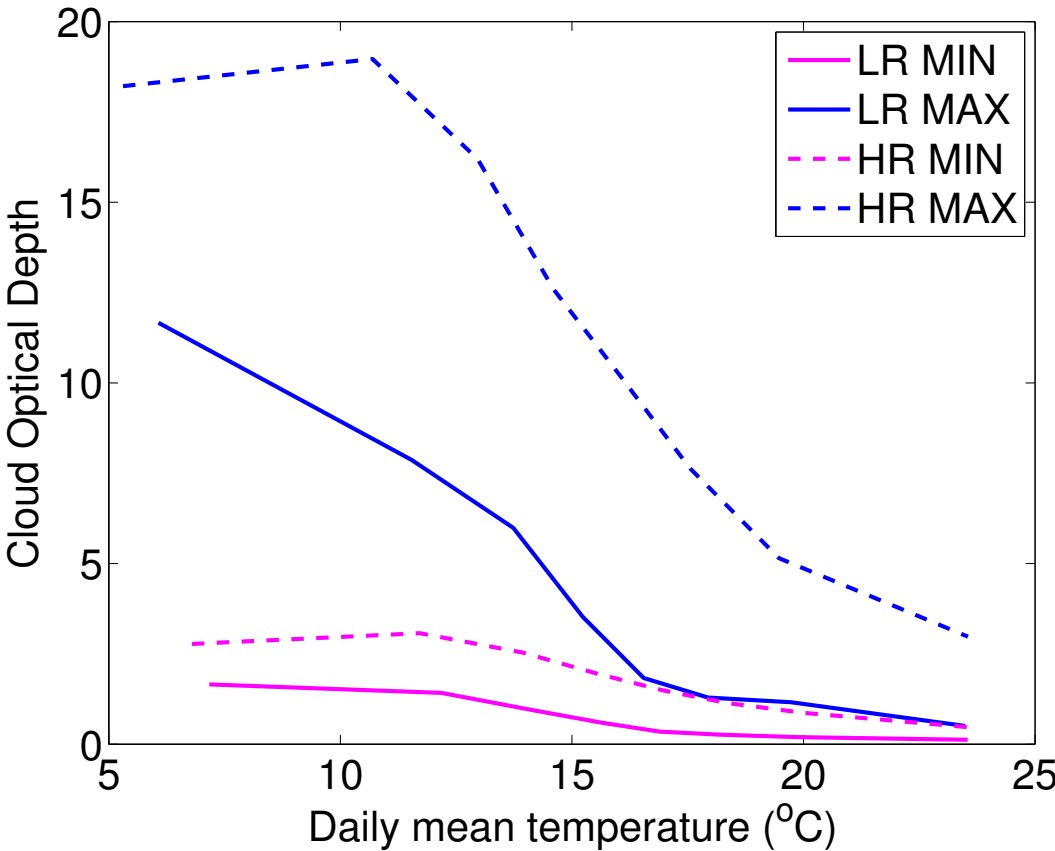

**Figure 5.** Hourly COD as a function of daily mean temperature for the LR MIN (full magenta line), LR MAX (dashed blue line), HR MIN (dashed magenta line) and HR MAX (dashed blue line) simulations.

with $\varepsilon$ corresponding to the precipitation efficiency, $Q$ the water vapor mixing ratio at the surface and $W$ the maximum vertical wind speed. This description is mostly valid for convective precipitation which result from a parcel that raises from the surface (Da Silva et al., 2018). Assuming the small changes of precipitation that we observe between the MAX and the MIN simulations, one can write :

$$\frac{Pr_{MAX} - Pr_{MIN}}{Pr_{MIN}} \approx \frac{\epsilon_{MAX} - \epsilon_{MIN}}{\epsilon_{MIN}} + \frac{Q_{MAX} - Q_{MIN}}{Q_{MIN}} + \frac{W_{MAX} - W_{MIN}}{W_{MIN}} \tag{3}$$

Figure 6 displays relative changes in convective precipitation vertical wind speed, precipitation efficiency, and surface water vapor mixing ratio between the LR MAX and LR MIN simulations for median and extreme precipitation. As expected from Fig. 5, the decrease of convective precipitation in the MAX simulation with respect to the MIN simulation tends to be weaker with increasing temperatures, from $-25\%$ at $10^{\circ}$C until almost $0\%$ at $22^{\circ}$C. Among the three factors that may impact the precipitation intensity, the vertical velocity seems to explain much of the reduction of convective precipitation. Indeed, among the $25\%$ of precipitation reduction at low temperatures, around $15\%$ are attributable to the weakening of vertical velocity in

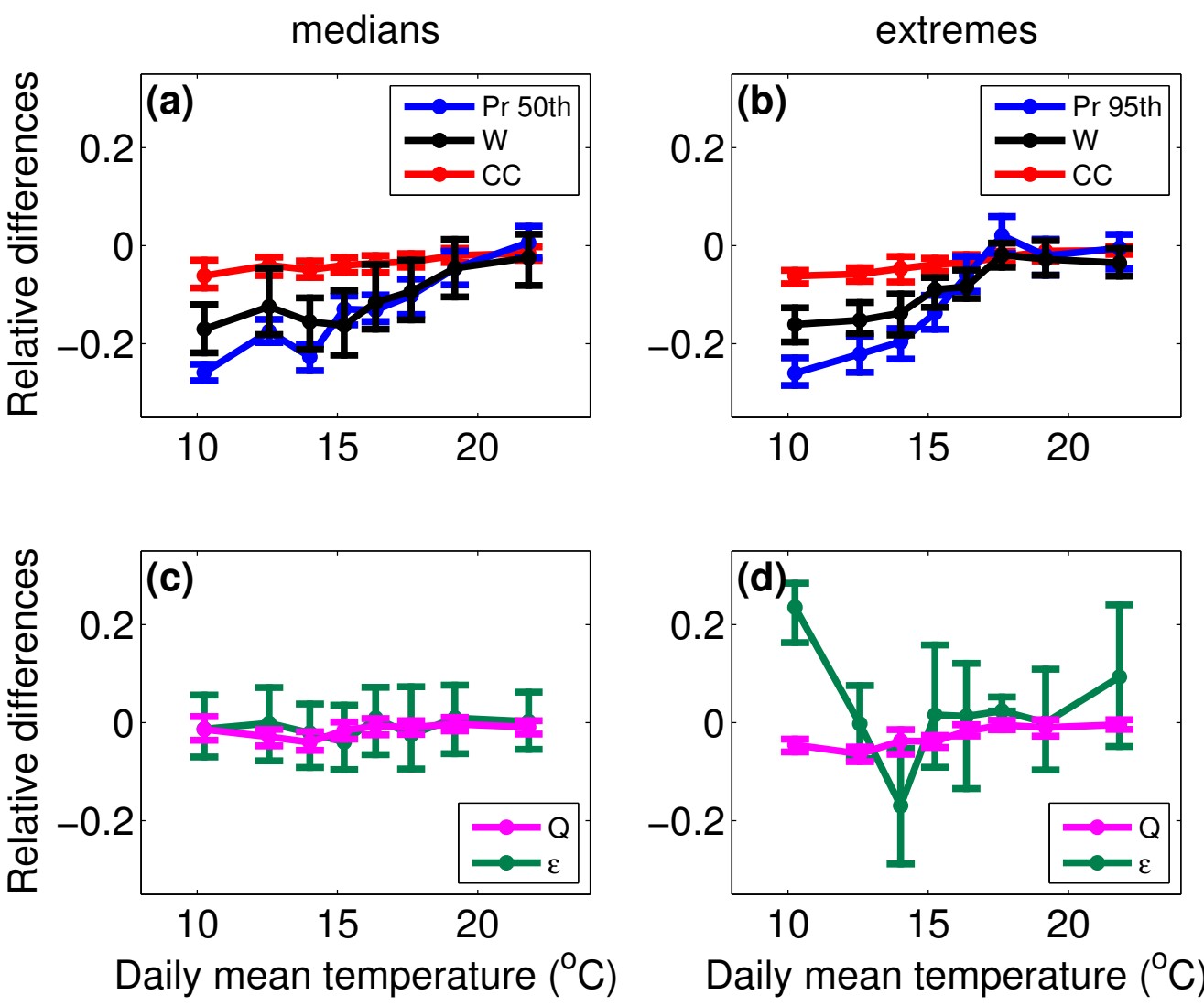

**Figure 6.** Relative differences between LR MAX and LR MIN simulations of convective precipitation (blue, a and b), vertical velocity (black, a and b), surface water vapor mixing ratio (magenta, c and d), precipitation efficiency (green, c and d) for median (a and c) and extreme (b and d) convective precipitation events as a function of the mean between the MIN and MAX daily mean temperature. The change expected according to the Clausius-Clapeyron law is displayed in red (a and b). Errorbars represent the 95 % confidence interval of the precipitation percentiles.

the MAX simulation. It is also striking in Fig. 6 that the variations of the difference of vertical velocity and of convective precipitation with temperature are perfectly similar, with stronger reductions for low temperature than for higher ones, while both precipitation efficiency and surface water vapor mixing ratio display insignificant or erratic variations with temperature. Indeed, the high variations of precipitation efficiency differences with temperature for precipitation extremes may not reflect a physical process but only the difficulty in retrieving precipitation efficiency from hourly outputs.

The fact that vertical velocity drives the changes in convective precipitation explains why the CC-scaling is completely inaccurate for predicting changes in convective precipitation by indirect effects. In fact, even the differences of surface water vapor mixing ratio between the MAX and MIN simulations do not exactly follow a CC-scaling due to increased relative humidity in the MAX simulation: while the CC law prediction is around $-4\%$, the reduction of surface water vapor mixing ratio in the MAX simulation is often less important. One would expect that the sub-CC scaling of surface water vapor mixing ratio differences would result in a sub-CC scaling of convective precipitation differences but it is actually the reverse (super-CC scaling) because of stronger changes in vertical velocity. Results are similar for both extreme and median precipitation except for precipitation efficiency differences which displays small variations for median precipitation and erratic variations for extreme precipitation which may not have a physical meaning. Indeed, differences in precipitation efficiency are constrained by the scaling of precipitation (eq. 3) and thus are expected to be negligible compared to the changes in vertical velocity. In our LR simulations, precipitation efficiency is calculated as a decreasing function of cloud base height and vertical wind shear. The increased surface relative humidity in the LR MAX simulation compared to the LR MIN simulation therefore would act in increasing the precipitation efficiency of the polluted simulation. The contribution of the change in the vertical wind shear is less evident and its high temporal and spatial variability (Markowski and Richardson, 2007) may explain errors in retrieving precipitation efficiency from hourly outputs.

Figure 7 is the same as Fig. 6 but for the HR total precipitation. Although the differences of vertical velocity and surface water vapor mixing ratio for median precipitation events have approximately the same behavior with temperature in the HR simulation with respect to the LR simulation, MAX-MIN differences of the median of total HR precipitation are stronger than the differences of the median of LR convective precipitation. Such positive bias compared to LR convective precipitation differences may be expected since Da Silva et al. (2018) showed that stratiform precipitation are increased in the MAX simulation. On the contrary it was found that hourly extreme precipitation are dominated by convective events at high temperatures (Loriaux et al., 2013). The decomposition of precipitation as a product of a thermodynamics, dynamics and a microphysics term made in the present study is theoretically better adapted to convective precipitation than to stratiform precipitation (Da Silva, 2018) and thus is not efficient in explaining differences of total median precipitation. In our LR simulations, we found that convective precipitation dominates extreme total precipitation from 10$^\circ$C (not shown), thus for most of our temperature bins. Therefore, the scaling of precipitation used in the present study (eq. 2) can be used for extreme total precipitation in the HR simulation. Differences of extreme total precipitation in the HR simulation are similar with the differences of extreme convective precipitation in the LR simulation and scale well with the differences of maximum vertical velocities.

In this set of simulations with explicit convection, changes in aerosol concentrations may have larger effects on convective precipitation efficiency, since aerosols directly interact with convective clouds. Such impact has moreover been stated in many

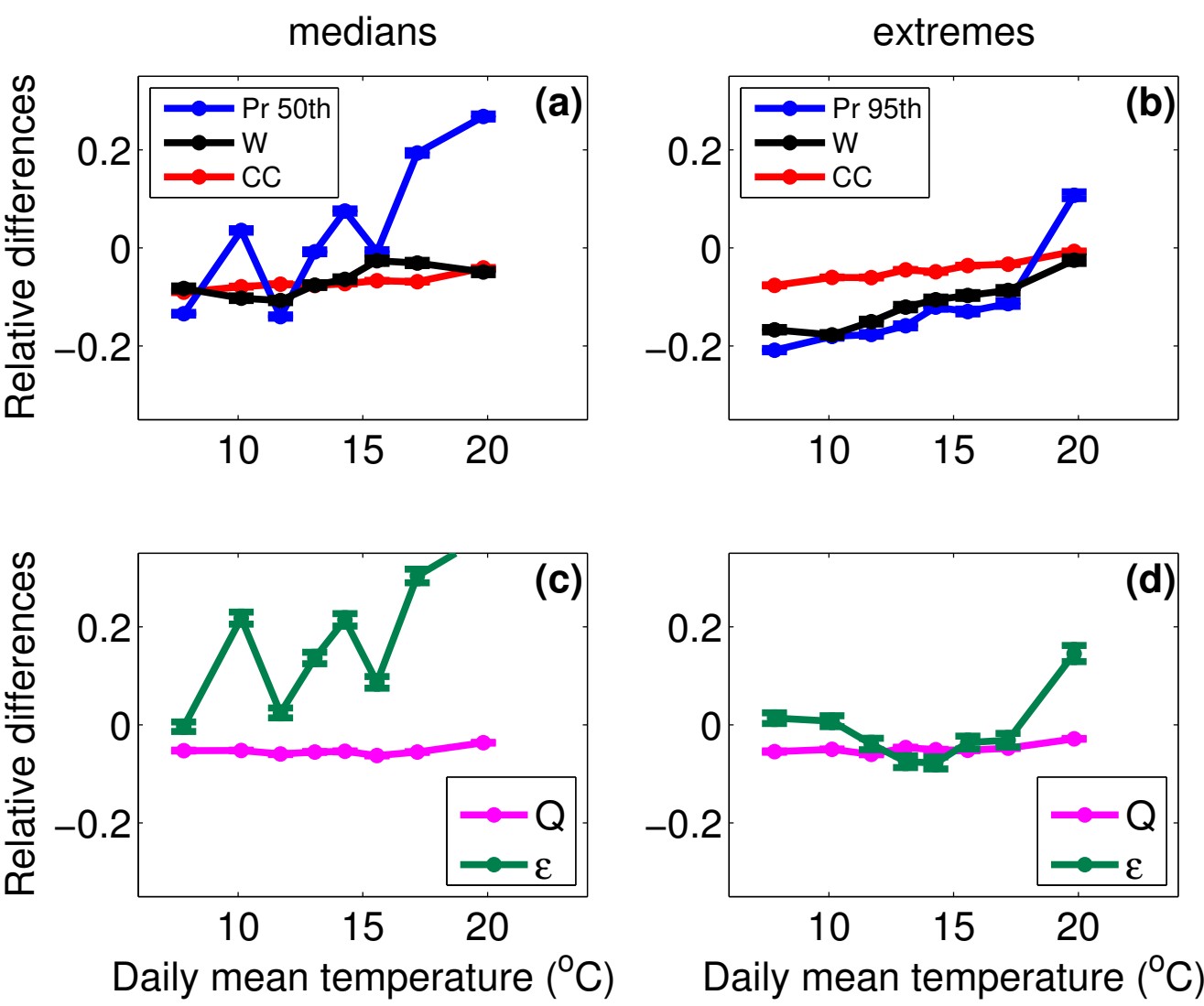

**Figure 7.** Relative differences between HR MAX and HR MIN simulations of total precipitation (blue) and vertical velocity (black, a and b), precipitation efficiency (green, c and d) and surface water vapor mixing ratio ($Q$, magenta, c and d) for median (a and c) and extreme (b and d) precipitation events as a function of the mean between the MIN and MAX daily mean temperature. The change expected according to the Clausius-Clapeyron law is displayed in red (a and b). Errorbars represent the 95 % confidence interval of the precipitation percentiles.

previous studies (Fan et al., 2009; Lebo and Seinfeld, 2011; Lebo and Morrison, 2014; Koren et al., 2015; Dagan et al., 2018; Miltenberger et al., 2018; Liu et al., 2019). The similarity of the precipitation differences with and without parameterized convection suggests that the changes of convective precipitation efficiency relatively small in the HR simulation. Precipitation efficiency was calculated only indirectly since it is not parameterized for explicitly resolved precipitation, by using the ratio of precipitation by the product of maximum vertical velocity and surface water vapor mixing ratio. Changes of precipitation efficiency between the MAX and the MIN simulations remain small in most of the temperature range. However, one can note a more significant increase of precipitation efficiency (around 15%) in the warmest temperature bin, associated with increased precipitation extremes in the MAX simulation. The exact nature of increased precipitation efficiency only at the highest temperatures is not obvious and remains further investigations which are beyond the scope of the present study. The use of hourly outputs may also not be adapted to analyse these particular extreme events at high temperatures which were shown to be shorter than extreme precipitation events at lower temperatures (Utsumi et al., 2011; Drobinski et al., 2016; Gao et al., 2018).

### 3.3 Contributions of humidity and temperature to stability changes

As mentioned in section 2.3, vertical velocity is calculated as the square root of CAPE. As seen in Fig.1, CAPE may be affected by both surface temperature and surface humidity. CAPE is calculated using the entire profile of temperature and relative humidity (RH). In this line, we want to quantify the contribution of both the temperature and RH profile changes in the decrease of CAPE in the MAX simulation. For that purpose we have substituted the vertical profile of temperature in the MIN simulation, by the vertical profile of temperature from the MAX simulation, and we have calculated two additional CAPEs, i.e. $CAPE_T$ (resp. $CAPE_{RH}$) calculated with the temperature profile from the MAX (resp. MIN) simulation and the relative humidity from the MIN (resp. MAX) simulation, as represented in Fig. 8. Using the 4 CAPEs ($CAPE_{MIN}$, $CAPE_{MAX}$, $CAPE_{RH}$ and $CAPE_T$) we can compute relative differences ($\Delta CAPE_{RH,1}$, $\Delta CAPE_{RH,2}$, $\Delta CAPE_{T,1}$, $\Delta CAPE_{T,2}$, and $\Delta CAPE$, see Fig. 8) and thus infer the contribution of temperature and RH vertical profiles in the change of CAPE between the MAX and the MIN simulations.

Figure 9 shows the total change of CAPE between the MAX and MIN simulations ($\Delta CAPE$), the RH contribution ($\Delta CAPE_{RH} = \frac{\Delta CAPE_{RH,1} + \Delta CAPE_{RH,2}}{2}$), and the temperature contribution ($\Delta CAPE_T = \frac{\Delta CAPE_{T,1} + \Delta CAPE_{T,2}}{2}$) as a function of daily mean temperature for median and extreme precipitation events. The quantity CAPE is lower in the MAX simulation with respect to the MIN simulation, and $\Delta CAPE$ is more negative at low temperatures (-30%) than at high temperatures (almost 0%). However one can see that $\Delta CAPE_T$ and $\Delta CAPE_{RH}$ have opposite signs. Indeed, the RH contribution is positive and decreases from about +40% at 10°C to about 0% at 22°C for median precipitation events. The fact that this contribution is positive is not a surprise since we have seen in Fig. 6 that the surface RH is higher in the MAX simulation. We can see that this apparently weak increase of RH in the MAX simulation has a strong effect on the CAPE at low temperatures. However the main contribution is negative and comes from the differences of vertical temperature profiles: values are ranging between -70% at low temperatures and -15% at high temperatures. Moreover, one can see similar variations of $\Delta CAPE$ and $\Delta CAPE_T$ with temperature.

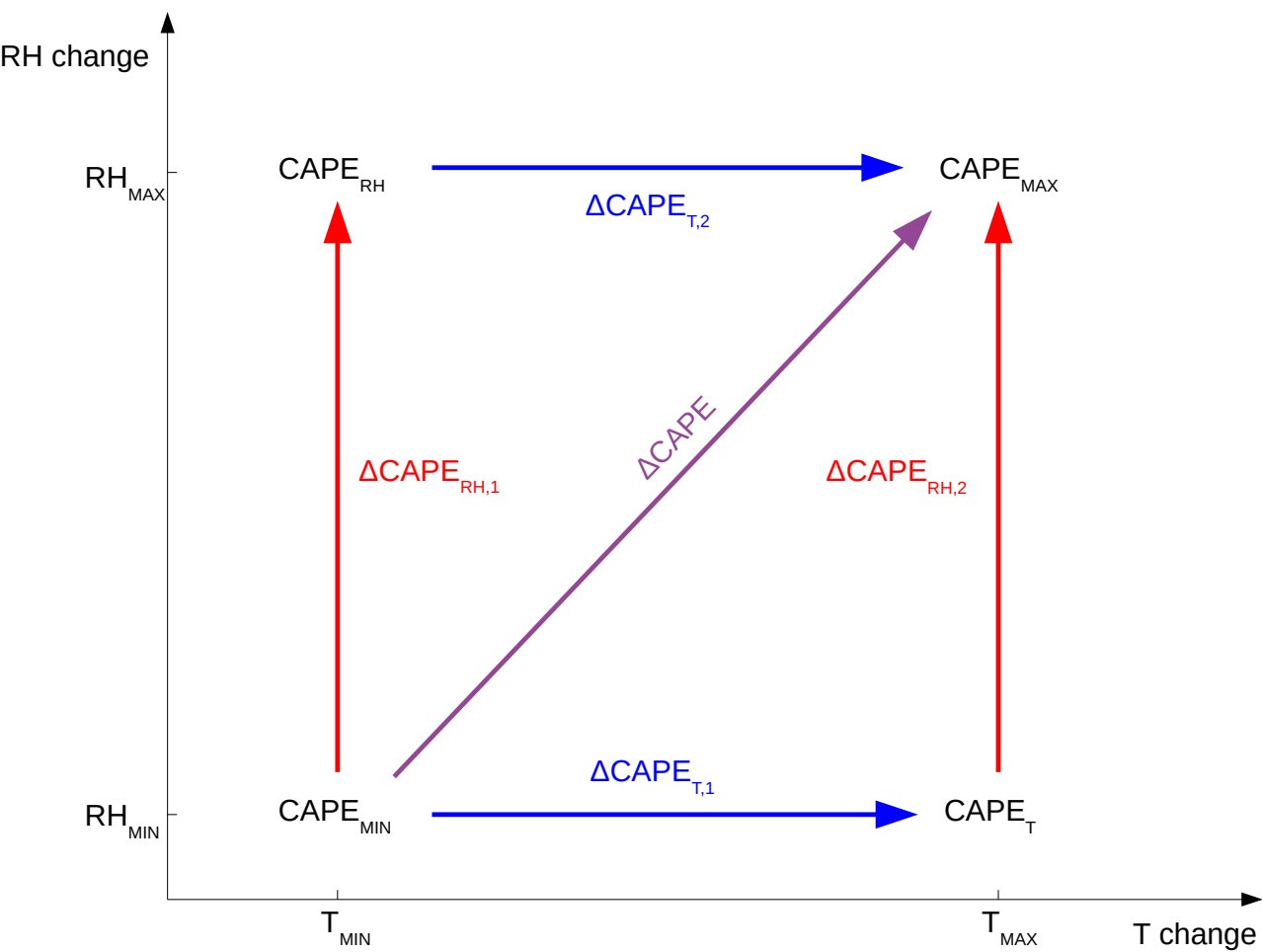

**Figure 8.** Schematic of the 2 possible CAPE differences that permit to evaluate the contribution of the vertical profile of temperature ($\Delta\text{CAPE}_{\text{T},1}$ and $\Delta\text{CAPE}_{\text{T},2}$) and the contribution of the vertical humidity profile ($\Delta\text{CAPE}_{\text{RH},1}$ and $\Delta\text{CAPE}_{\text{RH},2}$) to the change of total CAPE between the MAX and the MIN simulations ($\Delta\text{CAPE}$).

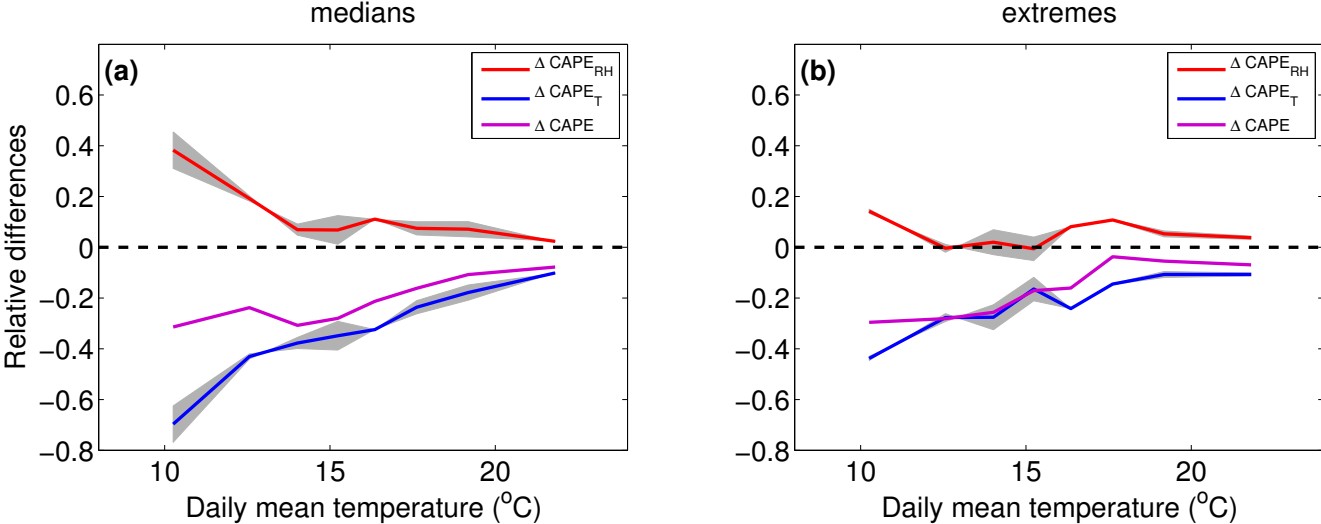

**Figure 9.** Relative differences of CAPEs for median (a) and extreme (b) convective precipitation events of the relative difference between the LR MAX and the LR MIN simulations (magenta, $\Delta$CAPE). The temperature contribution ($\Delta$CAPE$_T$) is displayed in blue and the relative humidity contribution ($\Delta$CAPE$_{RH}$) in red.

Figure 10 is the same as fig. 9 but for the HR simulations and total precipitation. The quantity $\Delta$CAPE is larger in the HR simulation with values that exceed -50% for a wide range of low temperatures in both median and extreme precipitation. These large values of $\Delta$CAPE result in small negative differences of maximum vertical wind speed that do not exceed -10% and are not correlated with total precipitation differences for median total precipitation events (see fig. 7) because of the coexistence of convective and stratiform events. For extreme events, which mostly consist of convective events, the discrepancy between the strong changes of CAPE and the weaker changes of vertical velocities between the HR MAX and the HR MIN simulation may be explained by enhanced release of latent heat at the freezing level caused by increased vertical mass transport of water droplets in polluted conditions, as suggested by previous studies (Khain et al., 2004, 2005; Rosenfeld et al., 2008; Lebo and Seinfeld, 2011; Fan et al., 2013). Indeed, in the present study, CAPE was calculated using a simple formula that does not account for differences in the load of rising parcels as long as their temperature and their relative humidity remain unchanged. According to the theory described in Rosenfeld et al. (2008), the aerosol concentrations of our MAX simulation may however be too high for invigorating updrafts but would instead weaken them. The reduced changes of vertical velocity between our HR MAX and HR MIN simulations may therefore have another origin. It is also expected that the diffusion efficiency increases when increasing aerosol loading since the resulting increase in cloud drop number would lead to an increase of the total surface area of cloud droplets, enhancing condensation and latent heat release (Pinsky et al., 2012). However, our simulations were done using a saturation adjustment scheme, which excludes the possibility of an increased cloud condensation (and its resulting stronger updraft) in the HR MAX simulation due to this process. Miltenberger et al. (2018) observed large increases of latent heating in the warm phase of clouds, thus not related with the theory of convective invigoration exposed in Rosenfeld et al.

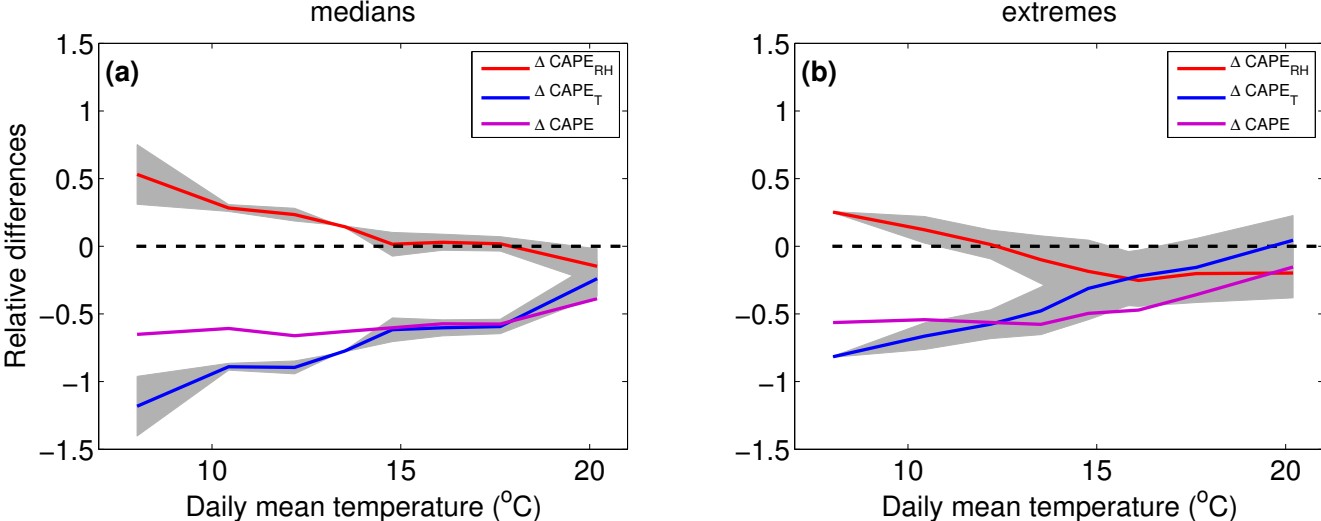

**Figure 10.** Relative differences of CAPEs for median (on the left) and extreme (on the right) precipitation events of the relative difference between the HR MAX and the HR MIN simulations (magenta, $\Delta$CAPE). The temperature vertical profile contribution ($\Delta$CAPE$_T$) is displayed in blue and the relative humidity vertical profile contribution ($\Delta$CAPE$_{RH}$) in red.

(2008). They attributed these changes to more organized cloud structures that limit dry air intrusions in the core of convective cells. Such an effect may also hold in our simulations since cloud cover and surface relative humidity are increased in the HR MAX simulation compared to the HR MIN simulation (not shown). According to the differences between the variations of CAPE and the variations of vertical velocities, this effect would limit the decrease of precipitation by about 10 % in the HR MAX simulation.

Otherwise contributions are similar to those of the LR simulations with mainly a positive contribution of RH and a strongly negative contribution from the temperature vertical profile.

The quantity CAPE is a non-linear function of the temperature and humidity profiles. Therefore, the change $\Delta$CAPE$_{T,1}$ is different from the change $\Delta$CAPE$_{T,2}$. Similarly, the change $\Delta$CAPE$_{RH,1}$ is different from the change $\Delta$CAPE$_{RH,2}$. The quantities $\Delta CAPE_{T,1}$ and $\Delta CAPE_{T,2}$ (resp. $\Delta CAPE_{RH,1}$ and $\Delta CAPE_{RH,2}$) delimit a grey area in Fig. 9 that represents

the uncertainty (relative to the non-linearity of CAPE) of the temperature (resp. RH) contribution. One can see that the effects of CAPE non-linearity are generally lower than the difference between each contribution. Where the grey areas do not intersect, i.e. in almost the entire temperature range for median precipitation, and for the cooler part of the distribution for extreme precipitation, comparison of $\Delta$CAPE$_T$, $\Delta$CAPE$_{RH}$ and $\Delta$CAPE strengthen the interpretation presented above: the negative value of $\Delta$CAPE can be attributed to temperature changes, pently buffered by RH changes.

However the vertical temperature profile can be changed in several ways, e.g. one can only change the vertical gradient of temperature or uniformly reduce the temperature on the vertical. In the first configuration the decrease of CAPE would be purely due to the increase of stability of the environment whereas in the second configuration the decrease of CAPE would be

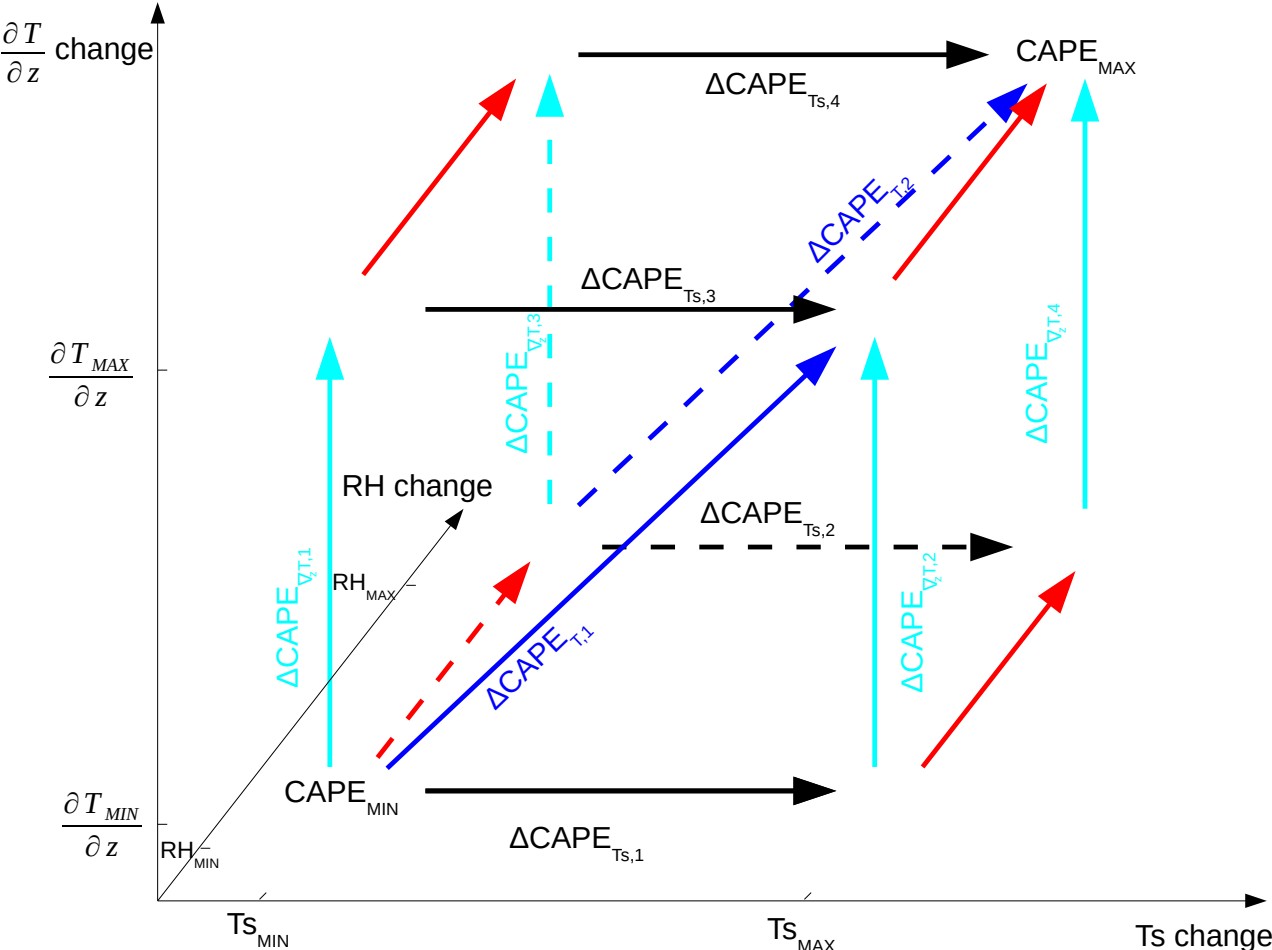

**Figure 11.** Schematic of the 4 possible CAPE differences that permit to evaluate the contribution of the vertical gradient of temperature ($\Delta CAPE_{\nabla_z T,1}$, $\Delta CAPE_{\nabla_z T,2}$, $\Delta CAPE_{\nabla_z T,3}$, and $\Delta CAPE_{\nabla_z T,4}$) and the contribution of the surface temperature ($\Delta CAPE_{Ts,1}$, $\Delta CAPE_{Ts,2}$, $\Delta CAPE_{Ts,3}$, and $\Delta CAPE_{Ts,4}$) to $\Delta$CAPE.

due to the surface air parcel temperature, more precisely to its reduced release of latent heat due to reduction of its initial water vapor content.

In this part, the temperature contribution is decomposed into two contributions, one from the vertical gradient of temperature and one from the surface temperature. The quantity CAPE can now be viewed as a function of three variables: the RH profile, the vertical temperature gradient and the surface temperature. As displayed in Fig. 11, for a given RH profile (from the MIN or the MAX simulation), we have substituted the vertical temperature gradient (resp. surface temperature) from the MIN simulation, by the vertical temperature gradient (resp. surface temperature) from the MAX simulation, and we have cal-
culated 4 additional CAPEs using the 4 new mixed profiles. By calculating relative differences of CAPE, one can evaluate

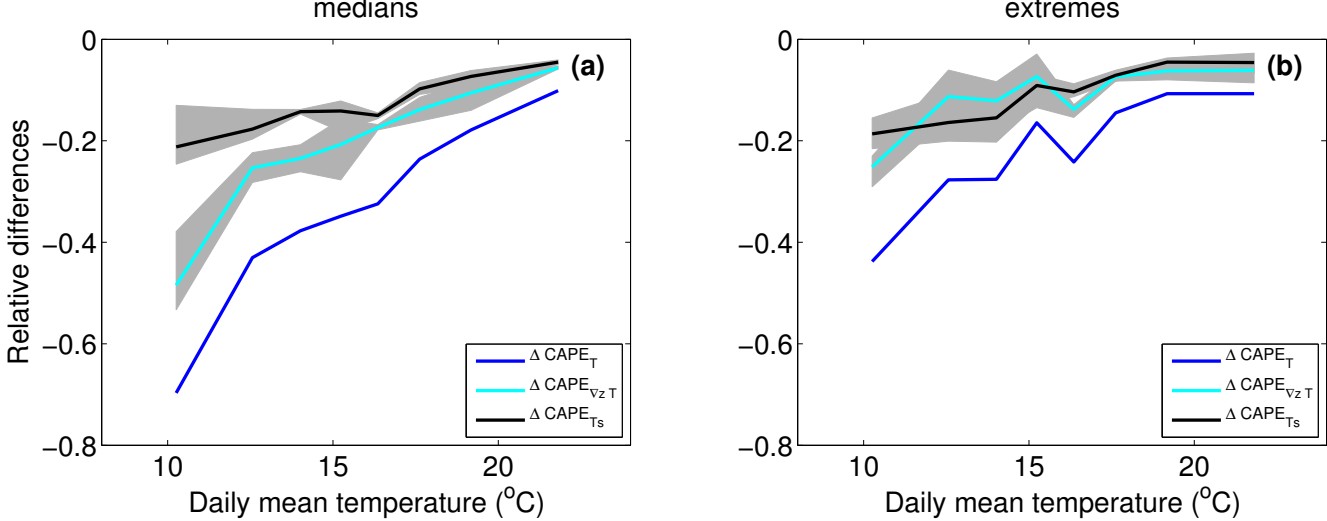

**Figure 12.** Relative differences of CAPEs for median (a) and extreme (b) convective precipitation events of the temperature contribution to the relative difference between the LR MAX and the LR MIN simulations (blue, $\Delta \text{CAPE}_T$). The surface temperature contribution ($\Delta \text{CAPE}_{Ts}$) is displayed in black and the temperature vertical gradient contribution ($\Delta \text{CAPE}_{\nabla_z T}$) in cyan.

the contribution of the surface temperature ($\Delta \text{CAPE}_{Ts} = \frac{1}{4} \sum_{i=1}^{i=4} \Delta \text{CAPE}_{Ts,i}$) and of the vertical gradient of temperature ($\Delta \text{CAPE}_{\nabla_z T} = \frac{1}{4} \sum_{i=1}^{i=4} \Delta \text{CAPE}_{\nabla_z T,i}$).

Figure 12 shows $\Delta \text{CAPE}_{\nabla_z T}$, $\Delta \text{CAPE}_{Ts}$ and $\Delta \text{CAPE}_T$ (as in Fig. 8) as a function of daily mean temperature for the LR simulations. The contribution of the vertical gradient of temperature and the contribution of the surface temperature are both

negative, indicating not only that the surface temperature is lower in the MAX simulation but also that this cooling is less important in the higher layers of the troposphere. Both processes tend to reduce the CAPE in the MAX simulation with respect to the MIN simulation. For median precipitation, the reduction of CAPE due to the vertical gradient of temperature (-10% at high temperatures to -50% at low temperatures) is more important than the reduction of CAPE due to the surface temperature (-10% at high temperatures to -20% at low temperatures). For extreme precipitation, contributions are similar and range between

-20% at low temperatures to -5% at high temperatures.

A similar analysis in the HR simulations is displayed in Fig.13. The results are very similar to those from the LR simulations with the exception that for extreme precipitation with low temperatures, the temperature gradient contribution is significantly larger than the surface temperature contribution.

The maximum and the minimum values of $\Delta \text{CAPE}_{Ts,i}$ (resp. $\Delta \text{CAPE}_{\nabla_z T,i}$) delimit a grey area in Figures 12 and 13 that

represent the uncertainty related to the CAPE non-linearity. It shows that for both HR and LR simulations, contributions are clearly different at low temperatures for median precipitation events whereas the uncertainty ranges tend to overlap at high temperatures. For extreme events, the non-linearity of SBCAPE does not permit to distinguish the two contributions for the entire range of temperatures of the LR simulations. In the HR simulations, the non-linearity uncertainty is also too large at

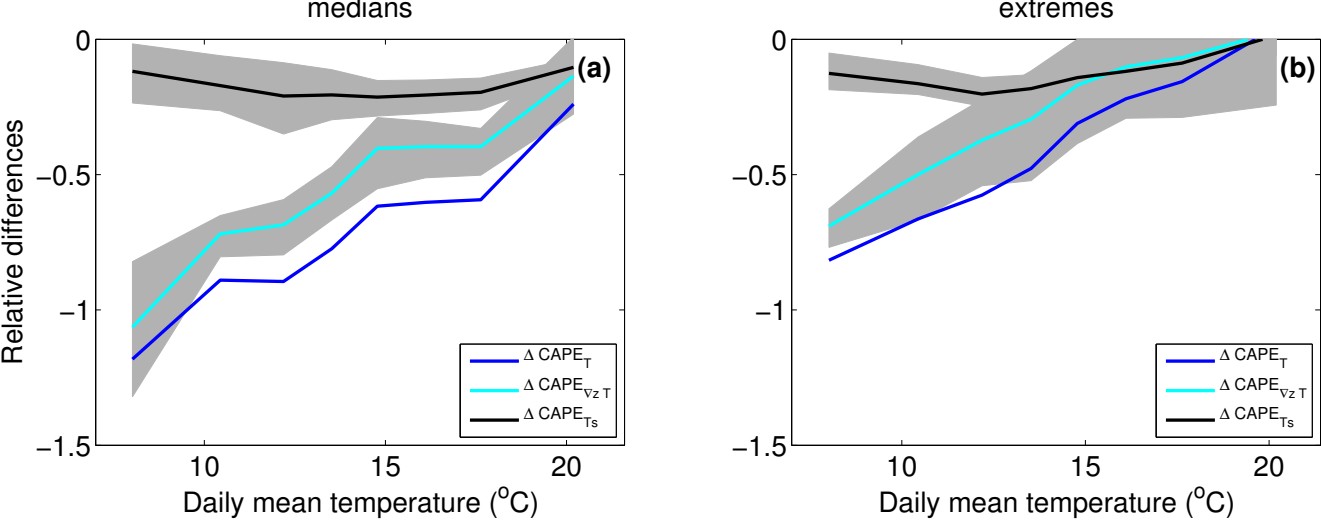

**Figure 13.** Relative differences of CAPEs for median (a) and extreme (b) precipitation events of the temperature contribution to the relative differences between the HR MAX and the HR MIN simulations (blue, $\Delta\text{CAPE}_\text{T}$). The surface temperature contribution ($\Delta\text{CAPE}_\text{Ts}$) is displayed in red and the temperature vertical gradient contribution ($\Delta\text{CAPE}_{\nabla_z\text{T}}$) in blue.

high temperatures to differentiate the two contributions. However the contribution of the vertical gradient of temperature is
significantly weaker than the contribution of the surface temperature at the lowest temperatures of the HR simulations.

## 4 Conclusions

An evaluation of the processes involved in the reduction of convective precipitation by aerosol indirect effects is performed in the present study in the frame of the temperature-precipitation relationship. Figure 14 summarizes the various involved processes and their qualitative contribution (size of the arrows). The temperature-precipitation approach permits to show that
aerosol indirect effects on convective precipitation are larger at low temperatures than at high temperatures because clouds are statically more frequent and optically thicker at cool temperatures in our area of interest. Da Silva et al. (2018) found that convective precipitation are weakened in polluted environment through reduced atmospheric instability and water availability. With a simple decomposition of the decrease of convective precipitation in the polluted simulation, we show that this decrease is dominated by differences in atmospheric stability rather than differences in the moisture content of air parcels (Fig. 14).
Therefore, the reduction of convective precipitation in the polluted simulation does not follow the Clausius-Clapeyron law: the simulated reduction in convective precipitation in a polluted environment compared to a pristine environment as determined in our simulations is actually stronger than the Clausius-Clapeyron scaling. Although taken into account in our simulations with explicit convection, the remaining aerosol indirect effects have only a relatively small impact on the precipitation efficiency of most of the extreme events in our simulations compared to the stability effect on convective updrafts. A noticeable increase in

precipitation efficiency was however detected at the highest temperatures in the polluted simulation. The exact nature of the associated increase of precipitation extreme is beyond the scope of this study and remains further investigations.

Using the CAPE parameter as a measure of the atmospheric stability, we perform an in-depth analysis that estimates the contribution of each variable to the weakening of convective updrafts in the polluted simulation. Quantifying uncertainties related to the non-linearity of the CAPE is essential to correctly attribute the contribution of each variable to the stability modifications. Our method gives a first estimation of these uncertainties and shows that they are small enough to assess the following conclusions. The weakening of vertical velocity in convective updrafts is essentially explained by the stabilisation of the vertical profile of temperature, which is partly compensated by an increase of relative humidity in the polluted simulation (Fig. 14). Our study also suggests the existence of a convective invigoration effect that also acts in compensating the stabilisation in the HR simulations. The origin of the invigoration could be linked to reduced dry air intrusions in the convective cores of our polluted simulation, as hypothesised in the study of Miltenberger et al. (2018). The modification of the vertical temperature gradient, due to a stronger cooling in the boundary layer than in the free troposphere in the polluted simulation, is the most important contribution for median precipitation events whereas for extreme precipitation it is of similar magnitude as the contribution of the surface temperature decrease. Our simulations performed at high resolution are consistent with these results even though their interpretation is made more difficult by the fact that convective and stratiform precipitation are melted together while having opposite responses to aerosol indirect effects (as seen in Da Silva et al., 2018).

Due to the poor understanding of aerosol indirect effects (Fan et al., 2016) and the known uncertainties associated with numerical modeling (Crétat et al., 2012; Diaconescu et al., 2007; Flaounas et al., 2011; Foley, 2010; Ramarohetra et al., 2015; Seth and Giorgi, 1998), aerosol indirect effects have been found to be sensitive to the model configuration (Seifert and Beheng, 2006; Lebo and Seinfeld, 2011; Lebo et al., 2012; Fan et al., 2012; Morrison, 2012; Lebo and Morrison, 2014; Hill et al., 2015; Johnson et al., 2015; White et al., 2017; Heikenfeld et al., 2019). The results found with our specific model configuration are uncertain in this sense and need to be confirmed by further numerical and observational studies. It is also worth noting that these results were obtained using extremely low and extremely high aerosol concentrations. While this approach permits to effectively retrieve the precipitation response to a drastic change in aerosol concentrations, it is uncertain that the magnitude of the involved processes does not change under less extreme aerosol conditions, as stated in other studies (Rosenfeld et al., 2008; Dagan et al., 2015; Miltenberger et al., 2018). Despite these limitations, our configuration highlights the importance of the background aerosol cloud radiative feedback and its repercussions on convective precipitation through changes of the thermodynamic profile of the atmosphere, often underestimated in case studies simulations. It is suggested in this study that this effect might be higher than any convective invigoration effect, as predicted in Fan et al. (2013) using shorter simulations on a smaller domain. A more realistic estimation of the aerosol indirect effects on convective precipitation could be carried out with the use of online-coupled models in which aerosol concentrations are evaluated with precise emission and transport schemes (Tuccella et al., 2019).

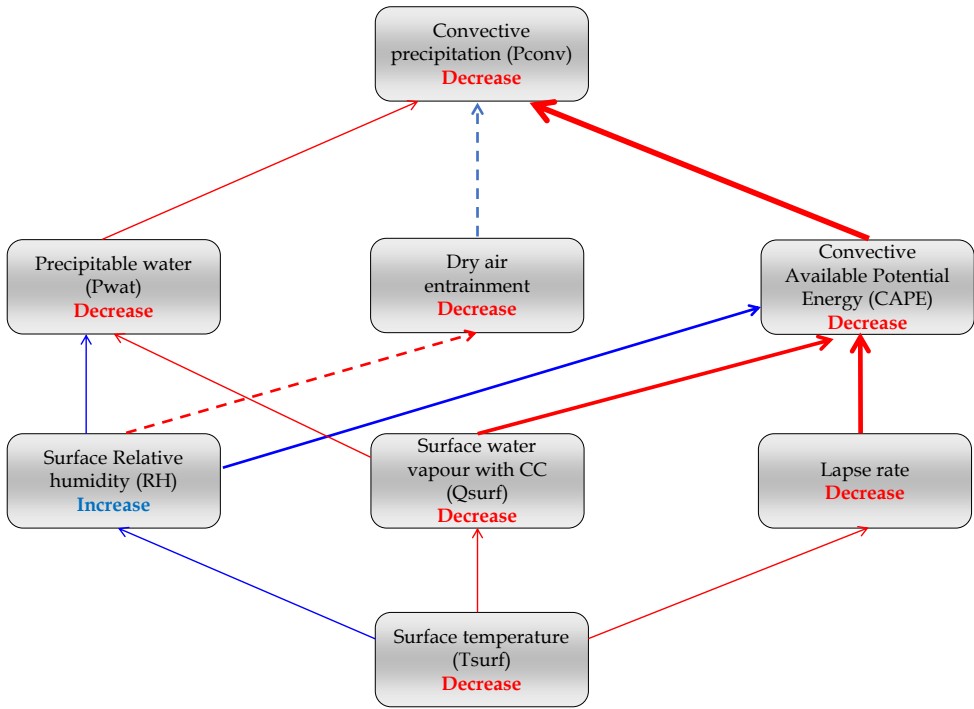

**Figure 14.** Detailed schematic summary of the causal sequence that links the decrease of surface temperature to the decrease of convective precipitation in our polluted simulation. The size of arrows gives a qualitative estimation of the contributions of each processes. Dashed arrows indicates uncertain paths.

*Data availability.* The WRF simulations used in this study can be obtained in the MISTRAL database website (registration required) at http://mistrals.sedoo.fr/?editDatsId=1503 or upon request to the authors.

*Author contributions.* The authors designed the numerical experiments. Nicolas Da Silva and Sylvain Mailler performed the simulations. Nicolas Da Silva prepared the manuscript with contributions from all co-authors.

*Competing interests.* The authors declare that they have no conflit of interest.

*Acknowledgements.* This work is a contribution to the HyMeX program (HYdrological cycle in the Mediterranean EXperiment) through INSU-MISTRALS support.

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
