# Peer review of "Aerosol indirect effects on the temperature-precipitation scaling"

_Atmospheric Chemistry and Physics, 2018_

## Short Comment (SC1) · 30 Jan 2019

Please correct the reference: Lenka, C. and Eva, H.: Simulated relationship between air temperature and precipitation over Europe: sensitivity to the choice of RCM and GCM, International Journal of Climatology, 38, 1595–1604, https://doi.org/10.1002/joc.5256, 2017

The correct reference is: Crhova, L. and Holtanova, E.: Simulated relationship between air temperature and precipitation over Europe: sensitivity to the choice of RCM and GCM, International Journal of Climatology, 38, 1595–1604, https://doi.org/10.1002/joc.5256, 2017

Thank you in advance! Eva Holtanova

---

## Referee Comment (RC1) · Anonymous Referee #1 · 26 Feb 2019

The authors present an investigation into the link between aerosol indirect effects (aerosol-cloud interactions) and convective precipitation, using a set of nested simulations with WRF. The study is part of an ongoing investigation by the author team into a topic of high relevance and broad interest. It is well designed, the analysis is adequately presented, and the results well supported. I have a number of minor comments and questions to clarify some issues, but see no major problems that should preclude publication in ACP. I therefore recommend publication subject to minor revisions, and thank the authors for an interesting manuscript.

My one potentially major comment, depending on the answer, is whether the aerosols included in this setup of WRF (and presented in section 2.1) include any amount of shortwave absorption? If they do, then the added heating rate through the atmospheric

column will also affect convection and stability (rapid adjustments, or the semi-direct effect), which might affect the results throughout the paper. If not, then this is not an issue - but it should still be noted. For a recent investigation of the rapid adjustments due to strongly absorbing aerosols (BC), see Stjern et al. JGRA 2017; this is potentially a very significant effect in some regions.

Minor comments:

* The abstract opens with "Indirect effects of aerosols were found to weaken..." Where? In the present manuscript, or in the previous litterature it builds on? (Both seem to be the case, but please clarify.)

* P1L18: "a hook shape". This term is used throughout the paper, but never fully explained. Please expand a bit, so the reader won't have to dig it out of the references.

* P2L21: Malavelle 2017, Nature Geoscience, should probably also be cited in this context.

* P3L29-30: How are the max and min values in WRF determined? Do they have any physical meaning, or are they simply the endpoints of the validity of some internal parametrization? This matters, because it affects how we should interpret the ranges found later in the study.

* P6L11: Have you tested that daily averaged temperature is indeed representative? How about days with strong diurnal cycle (which would be predominantly low-cloud conditions) vs weak (prevailing clouds), which could have the same average temperature but quite different convective precip event statistics?

* P6L20: most -> more?

* P6L20: Here and elsewhere, consider replacing "SBCAPE" with another term. It is not an intuitive abbreviation, nor short enough to function as a symbol. This becomes very clear on page 12 and in Figure 11, for instance. Why not just E_C, as the rest of the term is clear from the definition?

[Figure]

* Figures 3 and 4: Here I would have liked to see some ranges in addition to the lines. E.g. 25th-75th percentile for the medians, and 90th-99th for the extremes? This helps in interpreting the difference between the cases. Later figures have ranges shown, which makes them very clear.

* P9Eq3: This would be a partial derivative decomposition, I guess?

* P12L16: "Extreme precipitation are mostly of convective nature" -> add "events" and a reference, perhaps? (Or is it still Da Silva 2018? Not quite clear.)

* Figure 11: Again, this just illustrates the concept of partial derivatives... Perhaps this figure is overly complex? The point is made nicely by figure 8 already.

* Finally: This entire study is performed within WRF. That's OK, but I find little discussion of any possible limitations of that particular model. How broadly applicable do the authors think their results are? Are crucial elements still missing, even for WRF at such high resolution? (There is some discussion in the conclusions, but I would encourage expanding a bit on it.)

―――――――――――――――――――

---

## Referee Comment (RC2) · Anonymous Referee #2 · 11 Mar 2019

The paper uses a spring and summer of regional high resolution simulations over Europe to explore the link between precipitation, temperature and aerosol effects. A precipitation temperature-bin approach is used to order the data for high and low aerosol loadings.

I have a couple of main issues with the paper as it stands, outlined below. Until these main points are dealt with i do not think the paper is ready to be published in ACP.

Main points:

1. The results show a statistical relationship between precipitation and temperature, which is fine, but the subsequent comparison with a Clausius-Clapeyron expected increase needs some clarification or modification. The authors are conditionally sam-

pling on precipitation, but across two seasons, this will blend together different weather regimes together as the optical depth vs temperature plot indicates. It seems to me that what is happening here is that different dynamical regimes driven by large scale dynamics are being conflated with the surface temperature. If you isolated one case and increased the atmospheric temperature then i would expect to see the Clausius-Clapeyron-like behaviour, but the negative gradient suggests to me that changes in atmospheric stability driven by the global circulation is the main controller of the T-P relation. I think that this is not a useful comparison between the median observations and C-C as it stands.

I recommend that the data be reanalysed in a way that conditionally samples one type of convection (e.g. 'popcorn convection' only), perhaps by using cloud fraction thresholds. In some ways, the extremes analysis is doing the job of conditionally sampling on strong convection. By focusing on the most intense events the precipitation is probably linked to the strongest convection events in that temperature bin.

2. The title mentions indirect effects. In the introduction the first and second indirect effects are discussed, but observational evidence for the second indirect effect is felt to be inconclusive. That may be true, but the model used in this work does explicitly represent the second indirect effect through modification of the autoconversion process that will lead to reduced precipitation for increased aerosol, all things being equal. Given that one result of this analysis is that indirect effects appear less important than temperature changes it would be simple to confirm this in the model by running a senstivity test with the droplet number-autoconversion link disabled or fixed.

Other points:

p1 line 2-3. Indirect effects.... are these effects caused by increasing aerosol?

p1 line 8. Is this surface temperature or aloft?

p1 line 6-7. I don't follow this sentence. I thought that figure 3c, 4a showed that the

mean precipitation did not follow C-C?

p1 line 14. Can you explain more why the first guess is that the extremes are most likely to follow C-C? Is it because you are assuming that these are the most precipitation efficient events that can wring out all of the moisture from an ascending parcel? Can you argue against the means not following C-C?

p1 line 24. What percentiles characterise the extremes referred to here?

p2 line 3. Can you define what is meant by 'hook' shape? Is it anomalously high precipitation for the warmer temperatures compared to C-C?

p2 line 5. ...with respect to ... -> ...in constrast to... ?

p2 line 19-20. ...reduced droplet radius with increased aerosol concentrations for constant liquid water content.

p2 line 25-26 ...through a decrease in evaporation from the surface due... (could be confusion with droplet evaporation).

p2 line 20. Observations may be inconclusive but the model you are using explicitly links aerosol-> droplet number-> autoconversion.

p4 line 4. The MR configuration should also be introduced in this subsection.

p4 line 30. While recognising that this is a sensitivity test - a concentration of 10,000 cm-3 for ice is 1000-100000 times more than typically observed. This is likely to result in large extensive ice anvils that impact the radiative balance of the simulation. If the nudging timescale were longer than 6 hours this might become a problem. What do the cloud fields simulated look like when compared to observations? What does the precipitation time series look like for HR, LR and observed?

p5 line 6 - are these the MR mentioned in the figure 2 caption? Perhaps description should be included in section 2.1?

[Figure]

p6 line 22. How sensitive are the results to the use of values computed 1 hour earlier? How about 2 hours or 30 minutes?

p9 line 8 - 'at the surface' - is this truly at the surface or a screen level value (e.g. 1.5m)?

p12 line 1-2. I don't really follow this. Why should the C-C predict changes in convective precipitation due to indirect effects? Given that the model explicitly represents a suppression of autoconversion due to increased droplet number concentration (from increased aerosol number concentration) the change in precipitation efficiency, to first order, would seem to be more of a predictor of changes in precipitation due to aerosol effects.

p12 line 11-19. This discussion ignores the fact that the microphysical scheme has autoconversion, and related processes, that is directly affected by the number concentration of aerosol and hence droplets. The HR can represent these effects explicitly in the convective clouds whereas the parameterised convection in the LR configuration will not represent aerosol effects. The assertions made here could be tested by disabling the link between droplet number and autoconversion in a sensitivity test of the HR configuration.

p19 conclusions. Figure 14 has no links to changes in microphysical processes directly affected by changes in aerosol. This may be true of the real world, but as far as i can see this was not cleanly demonstrated with the model (see comment about p12 line 11-19).

---

## Author Comment (AC1) · 7 May 2019

**Aerosol indirect effects on temperature-precipitation scaling**
*By N. Da Silva, S. Mailler and Ph. Drobinski*

Response to the comments of Anonymous Referee #1

Dear Anonymous Referee #1,

We are grateful for your careful reading of our manuscript and for pointing out points such as the fact that precipitation extremes were dominated by convective precipitation. We have taken into account all of your suggestions and modified the text accordingly. In the text below, your comments are in italics, our answers in straight black fonts, and the text in blue describes (and generally reproduces) the changes that have been brought to the manuscript.

The Authors.

*My one potentially major comment, depending on the answer, is whether the aerosols included in this setup of WRF (and presented in section 2.1) include any amount of shortwave absorption? If they do, then the added heating rate through the atmospheric will also affect convection and stability (rapid adjustments, or the semi-direct effect), which might affect the results throughout the paper. If not, then this is not an issue - but it should still be noted. For a recent investigation of the rapid adjustments due to strongly absorbing aerosols (BC), see Stjern et al. JGRA 2017; this is potentially a very significant effect in some regions.*

We used 2 different aerosol climatologies in this setup, one for the microphysical scheme and one for the radiative scheme. In our sensitivity experiment, only the climatology of the microphysical scheme is modified. Therefore, aerosols do include shortwave absorption, but this absorption is identical in both simulations since the aerosol radiative climatology is the same.

This explanation is in the text on line 16 of page 4:
« Another climatology of aerosols from Tegen et al. (1997) is used in this radiative scheme and therefore is not affected by any changes in the microphysical aerosol climatology, which enables us to perform sensitivity experiments of the indirect effects of aerosols
with fixed aerosol direct effect. »

A reference to the study of Stjern et al. (2017) has been added in the introduction.
«They have shown that the consecutive surface cooling not only reduces the water content but also stabilizes the atmosphere as suggested by Fan et al. (2013); Morrison and Grabowski 2011; Stjern et al. (2017) ...»

*The abstract opens with "Indirect effects of aerosols were found to weaken..." Where? In the present manuscript, or in the previous litterature it builds on? (Both seem to be the case, but please clarify.)*

It refers to the previous literature, and especially the Da Silva et al. (2018) study, for which the present manuscript can be seen as a follow-up. The first sentence of the abstract has been modified in order to be explicit:
«Convective precipitation are known to be negatively affected by aerosol indirect effects through reduced precipitable water and convective instability, as stated in the previous literature.»

*P1L18: "a hook shape". This term is used throughout the paper, but never fully explained. Please expand a bit, so the reader won't have to dig it out of the references.*

This term has been explicited in the new version of the manuscript:
«Although less documented than extremes, a "hook shape" of the temperature-precipitation relationship, that is a positive slope at low temperatures and a negative slope at high temperatures, is also suggested for mean precipitation (Zhao and Khalil, 1993; Madden and Williams, 1978; Crhova and Holtanova, 2017; Rodrigo, 2018) as well as differences between land and sea areas (Adler et al., 2008; Trenberth and Shea, 2005).»

*P2L21: Malavelle 2017, Nature Geoscience, should probably also be cited in this context.*

This citation was added in the manuscript.

*P3L29-30: How are the max and min values in WRF determined? Do they have any physical meaning, or are they simply the endpoints of the validity of some internal parametrization? This matters, because it affects how we should interpret the ranges found later in the study.*

These values maximize the potential effect of aerosols and correspond to the lowest and highest values that the microphysical parameterization tolerates. They are too extreme compared to observations and therefore do not have any physical meaning. The ranges found later in the study should be interpreted as upper bounds. The following sentence has been added in the 'simulation experiment' section: « It is however important to keep in mind that the ranges that will be found in this study should be interpreted as an upper bound of aerosol indirect effects. »

*P6L11: Have you tested that daily averaged temperature is indeed representative? How about days with strong diurnal cycle (which would be predominantly low-cloud conditions) vs weak (prevailing clouds), which could have the same average temperature but quite different convective precip event statistics?*

The choice of the daily averaged temperature has also been done for consistency with previous literature. However we have not tested if this temperature is indeed representative of the air mass. Days with strong diurnal cycle (sunny days) versus days with weak diurnal cycle (cloudy days) are indeed confused in our analysis. We think that the daily averaged value is not perfect, but might be more representative of the airmass than an instantaneous value which is might be affected by rain for example.

*P6L20: most -> more?*
It has been corrected in the new version of the manuscript.

*P6L20: Here and elsewhere, consider replacing "SBCAPE" with another term. It is not an intuitive abbreviation, nor short enough to function as a symbol. This becomes very clear on page 12 and in Figure 11, for instance. Why not just $E\_C$, as the rest of the term is clear from the definition?*

SBCAPE is the usual abbreviation to state for the convective energy that would acquire a parcel that is raised from the surface. It is used by several prediction centers like the Storm Prediction Center and the European Severe Storm Laboratory. It might be subjective, but we think that SBCAPE is

more intuitive than E_C which is less used in the literature. However, we admit that this term is a bit long to be used in our study. Since the fact that it is a 'surface-based' CAPE is not discussed in our article (with respect to other possible CAPE calculations such as 'Most Unstable' CAPE), we propose to remove SB and keep CAPE. It has been modified in the new version of the manuscript.

*Figures 3 and 4: Here I would have liked to see some ranges in addition to the lines. E.g. 25th-75th percentile for the medians, and 90th-99th for the extremes? This helps in interpreting the difference between the cases. Later figures have ranges shown, which makes them very clear.*

The new version of the manuscript include ranges for these figures. The ranges chosen are the 95% confidence intervals, as in figure 6 and 7. Descriptions of these errobars have been added in the caption of the corresponding figures.

*P9Eq3: This would be a partial derivative decomposition, I guess?*

It is indeed a partial derivative decomposition done from the logarithm of Eq. 2:

$$\ln(Pr) = \ln(\varepsilon) + \ln(W) + \ln(Q) + C$$

where C is a constant.

$$d(\ln(Pr)) = d(\ln(\varepsilon)) + d(\ln(W)) + d(\ln(Q))$$

which gives:

$$\frac{dPr}{Pr} = \frac{d\varepsilon}{\varepsilon} + \frac{dW}{W} + \frac{dQ}{Q}$$

Assuming small changes between both simulations, one can write the approximate equation by replacing infinitesimal differences (d) by differences between both simulations (Δ):

$$\frac{\Delta Pr}{Pr} \approx \frac{\Delta \varepsilon}{\varepsilon} + \frac{\Delta W}{W} + \frac{\Delta Q}{Q}$$

In the manuscript, the equal sign has been replaced by an approximately equal sign.

*P12L16: "Extreme precipitation are mostly of convective nature" -> add "events" and a reference, perhaps? (Or is it still Da Silva 2018? Not quite clear.)*

This sentence was written in order to explain why the scaling seems to work better for extreme total precipitation than in median total precipitation. The argument is that the proportion of convective precipitation is higher in total extreme precipitation than in total median precipitation. Since the scaling (Pr = epsilon W Q) is more adapted to convective precipitation, one can expect a better fit for extreme total precipitation than for extreme median precipitation, which is actually the case. Loriaux et al. (2013) stated that "On an hourly time-scale, precipitation extremes are predominantly stratiform at low temperatures, while at high temperatures convective extremes become dominant." In our LR simulations, we found that convective precipitation start to dominate precipitation extremes from 10°C (not shown), which corresponds to almost the whole range of temperature in our case. On the contrary, we found that stratiform precipitation dominate median precipitation for the whole range of temperature. It is therefore conform with our explanation.

*We added a short explanation and a reference to the work of Loriaux et al. (2013).*

*\* Figure 11: Again, this just illustrates the concept of partial derivatives... Perhaps this figure is overly complex? The point is made nicely by figure 8 already*

It indeed illustrates the concept of partial derivative. This figure was done to introduce variables that are discussed in the text, so that the text is easier to read and the reader can refer to this figure if needed.

*\* Finally: This entire study is performed within WRF. That's OK, but I find little discussion of any possible limitations of that particular model. How broadly applicable do the authors think their results are? Are crucial elements still missing, even for WRF at such high resolution? (There is some discussion in the conclusions, but I would encourage expanding a bit on it.)*

We expanded a bit the conclusion in order to reveal the main limitations of our model setting. A deeper discussion is made on the previous companion paper (Da Silva et al., 2018).

---

## Author Comment (AC2) · 7 May 2019

**Aerosol indirect effects on temperature-precipitation scaling**
*By N. Da Silva, S. Mailler and Ph. Drobinski*

Response to the comments of Anonymous Referee #2

Dear Anonymous Referee #2,

We are grateful for your careful reading of our manuscript and for pointing out points such as the second indirect effect. We have taken into account all of your suggestions and modified the text accordingly. In the text below, your comments are in italics, our answers in straight black fonts, and the text in blue describes (and generally reproduces) the changes that have been brought to the manuscript.

The Authors.

*1. The results show a statistical relationship between precipitation and temperature, which is fine, but the subsequent comparison with a Clausius-Clapeyron expected increase needs some clarification or modification. The authors are conditionally sampling on precipitation, but across two seasons, this will blend together different weather regimes together as the optical depth vs temperature plot indicates. It seems to me that what is happening here is that different dynamical regimes driven by large scale dynamics are being conflated with the surface temperature. If you isolated one case and increased the atmospheric temperature then i would expect to see the ClausiusClapeyron-like behaviour, but the negative gradient suggests to me that changes in atmospheric stability driven by the global circulation is the main controller of the T-P relation. I think that this is not a useful comparison between the median observations and C-C as it stands.*

We do not deny any contribution from large scale dynamics changes in our temperature-precipitation relationship which is indeed spread over several seasons as many other studies of the temperature-precipitation relationship (Lenderink, 2008; Hardwick, 2010; Utsumi, 2011; Drobinski, 2016). The Clausius-Clapeyron scaling is a proxy of the expected precipitation change with constant weather regimes, relative humidity and precipitation efficiency. Comparisons with the CC-law, thus inform us on the validity of the latter hypotheses accross the temperature range covered by these 2 seasons. We found that the CC scaling is quite similar to the scaling of median convective precipitation even across two seasons. The explanation of this scaling is not the main topic of the article, that is why it is not investigated. For median total precipitation, we found a negative slope. Indeed, although not specified in our article, it is most likely due to changes in large scale dynamics between spring season (cool temperatures) and summer season (warmer temperatures).

We believe that adding some possible explanation of the scalings that we observe would clarify the text. It has been done modifying the first paragraph of the result section.

*I recommend that the data be reanalysed in a way that conditionally samples one type of convection (e.g. 'popcorn convection' only), perhaps by using cloud fraction thresholds. In some ways, the extremes analysis is doing the job of conditionally sampling on strong convection. By focusing on the most intense events the precipitation is probably linked to the strongest convection events in that temperature bin.*

If the goal of the suggested reanalysis is to select events with similar large scale dynamics, it does

not appear to be necessary in the scope of our article which is more focused on the study of precipitation differences between each simulation. The question of convection type impact on the temperature-precipitation scaling is obviously an interesting question, but it needs to be treated in an entire study.

*2. The title mentions indirect effects. In the introduction the first and second indirect effects are discussed, but observational evidence for the second indirect effect is felt to be inconclusive. That may be true, but the model used in this work does explicitly represent the second indirect effect through modification of the autoconversion process that will lead to reduced precipitation for increased aerosol, all things being equal. Given that one result of this analysis is that indirect effects appear less important than temperature changes it would be simple to confirm this in the model by running a sensitivity test with the droplet number-autoconversion link disabled or fixed.*

This is indeed a point that seems to be missing in our study, but several issues forced us not to do these additional simulations. The «all things being equal» assertion would not be respected when running such a sensitivity test. In particular, when one would modify the autoconversion rate it would also have an effect on the averaged mass of precipitating clouds and then on cloud albedo. Thus, accelerating the autoconversion rate in the MAX simulation may as well reduce the radiative background effect in such 6-months simulations. A potential solution was to initiate each day of this new MAX simulation by the old MAX simulation, considering that the radiative effect is an effect that forms only trough several days of simulation (which might not be totally true). The problem of such a setup is that the same water may be precipitated several times in the simulation, which would overestimate the second indirect effect.

In our study, the precipitation extreme scaling budget clearly shows that changes in convective/extreme precipitation are similar to changes in vertical velocity and are not very sensitive to changes in thermodynamics, which indirectly discard an important effect of precipitation efficiency. Another argument is the similarity of the curves (fig. 6b vs fig. 7b) with and without convective parameterization.

*Other points:*
*p1 line 2-3. Indirect effects.... are these effects caused by increasing aerosol?*

These effects are indeed caused by an increase in aerosol concentrations used in the microphysics scheme.

*p1 line 8. Is this surface temperature or aloft?*

It is precisely the temperature at the first vertical grid level. The term « surface » has been added in the abstract of the new version of the manuscript.

*p1 line 6-7. I don't follow this sentence. I thought that figure 3c, 4a showed that the mean precipitation did not follow C-C?*

Indeed, as shown in figure 3c and 4a, mean precipitation does not follow the C-C law. However the abstract mentions convective precipitation, which are displayed in figure 3a (medians) and 3b (extremes), and which do follow the C-C law.

*p1 line 14. Can you explain more why the first guess is that the extremes are most likely to follow C-C? Is it because you are assuming that these are the most precipitation efficient events that can wring out all of the moisture from an ascending parcel? Can you argue against the means not following C-C?*

It is expected that extremes would follow the CC law since extremes are supposed to remove all the vapour content of the atmosphere. On the other hand, mean precipitation are constrained by an energetic budget between atmospheric radiative cooling and surface sensible and latent fluxes (Allen and Ingram 2002; Held and Soden 2006; Muller and O'Gorman 2011; Muller et al 2013). As a result, the increase of mean precipitation is expected to be lower than the one of precipitation extremes with temperature. This discussion has been added in the first paragraph.

*p1 line 24. What percentiles characterise the extremes referred to here?*

99th and 99.9th for hourly precipitation and only 99.9th for daily precipitation.
It has been added between parentheses in the text.

*p2 line 3. Can you define what is meant by 'hook' shape? Is it anomalously high precipitation for the warmer temperatures compared to C-C?*

'Hook shape' has the same meaning than in the Drobinski et al. (2016) study. It refers to the shape of the temperature-precipitation scaling which displays an increase slope at low temperatures, a precipitation peak at middle temperatures, and a negative or weaker slope at high temperatures. The definition has been added in the text.

*p2 line 5. ...with respect to ... -> ...in constrast to... ?*

The text has been modified accordingly.

*p2 line 19-20. ...reduced droplet radius with increased aerosol concentrations for constant liquid water content.*

The text has been modified accordingly.

*p2 line 25-26 ...through a decrease in evaporation from the surface due... (could be confusion with droplet evaporation)*

The text has been modified accordingly.

p2 line 20. Observations may be inconclusive but the model you are using explicitly links aerosol-> droplet number-> autoconversion.

The second aerosol indirect effect is discussed in the core and in the conclusion of the new manuscript (cf other points).

*p4 line 4. The MR configuration should also be introduced in this subsection.*

The text has been modified accordingly.

*p4 line 30. While recognising that this is a sensitivity test - a concentration of 10,000 cm-3 for ice is 1000-100000 times more than typically observed. This is likely to result in large extensive ice anvils that impact the radiative balance of the simulation. If the nudging timescale were longer than 6 hours this might become a problem. What do the cloud fields simulated look like when compared to observations? What does the precipitation time series look like for HR, LR and observed?*

Extreme aerosol concentrations were taken to avoid noise, that we observed at first when realizing a pair of simulation with a factor 2 in aerosol concentrations. As indicated in the table of Da Silva et al. (2018) study, there are some unrealistic values in terms of drop number, liquid water content, cloud optical depth. But these values only affect a little the radiative budget at the surface (while being sufficient to decrease surface temperatures by 0,5 K in the MAX simulation). Thus the extreme change of aerosol concentrations does not result in a drastic change in the radiative balance of the simulations. The reason is that the parameterization of cloud condensation only depends on sursaturation and not on aerosol concentration. It means that an increase in aerosol concentration does not explicitly favors condensation. It leads to very thick anvils but not necessarily larger. The precipitation time serie of one grid point for both HR and LR simulations is also shown in the Da Silva et al. (2018) study and seems realistic.

*p5 line 6 - are these the MR mentioned in the figure 2 caption? Perhaps description should be included in section 2.1?*

This sentence indeed refers to MR simulations. The meaning of MR has been added to the manuscript.

*p6 line 22. How sensitive are the results to the use of values computed 1 hour earlier? How about 2 hours or 30 minutes?*

Results were found similar when using values computed at the same hour or 2 hours before, with sometimes higher variability. The 1 hour earlier was chosen arbitrary, and considering that the output frequency is hourly.

*p9 line 8 - 'at the surface' - is this truly at the surface or a screen level value (e.g. 1.5m)?*

It is taken at the first grid vertical level that is around 28 m above the surface for oceans. For lisibility, we used 'surface' to designate this level. A parenthesis has been added in the method section : « (centered around 28 m above the ground, hereafter referred to as surface) »

*p12 line 1-2. I don't really follow this. Why should the C-C predict changes in convective precipitation due to indirect effects? Given that the model explicitly represents a suppression of autoconversion due to increased droplet number concentration (from increased aerosol number concentration) the change in precipitation efficiency, to first order, would seem to be more of a predictor of changes in precipitation due to aerosol effects.*

Indirect effects were found to reduce surface temperatures and thus water vapor availibility for precipitation. In the hypothesis of constant relative humidity, convective instability and precipitation efficiency, the change of convective precipitation would be similar to the one expected by the CC law.

Instead, the change in surface temperature is accompanied by a change in convective instability which has much more impact. For precipitation efficiency, note that it can only be impacted indirectly in the LR simulation, since the convection scheme does not explicitely take into account aerosol concentrations. In the HR simulation, precipitation efficiency, while hard to evaluate, is indeed an explicit function of aerosol concentrations. However for extremes, we can see that the relative changes in precipitation are very similar to the one of vertical velocity, suggesting that the contribution of precipitation efficiency is also low.

*p12 line 11-19. This discussion ignores the fact that the microphysical scheme has autoconversion, and related processes, that is directly affected by the number concentration of aerosol and hence droplets. The HR can represent these effects explicitly in the convective clouds whereas the parameterised convection in the LR configuration will not represent aerosol effects. The assertions made here could be tested by disabling the link between droplet number and autoconversion in a sensitivity test of the HR configuration*

A discussion was added at the end of this paragraph with justification on discarding the second indirect effect of aerosol (as stated above).

*p19 conclusions. Figure 14 has no links to changes in microphysical processes directly affected by changes in aerosol. This may be true of the real world, but as far as i can see this was not cleanly demonstrated with the model (see comment about p12 line 11-19).*

Figure 14 is a snapshot of the bigger scheme presented in figure 1. The conclusion has been enlarged to discuss the second indirect effect.

---

## Author Comment (AC3) · 7 May 2019

Dear Eva Holtanova,

We apology for this mistake which has been corrected in the new version of the manuscript.

The Authors.

---

## Author Response (AR2)

**Aerosol indirect effects on temperature-precipitation scaling**
*By N. Da Silva, S. Mailler and Ph. Drobinski*

Dear ACP Editor,

Thank you for your careful reading of our manuscript and for pointing out that a discussion of other aerosol indirect effects on convective clouds was missing in our manuscript. We have taken into account all of your suggestions and modified the text accordingly. In the text below, your comments are in italics, our answers in straight black fonts, and the text in blue describes (and generally reproduces) the changes that have been brought to the manuscript.

We hope that with these changes our study will be retained for publication in ACP.

Kind Regards,

The Authors.

*Comments to the Author:*

*Dear Authors*

*Thank you for providing the revised manuscript "Aerosol indirect effects on temperature-precipitation scaling" in response to the request for revisions.*

*Previous marks given by the reviewers were not sufficient to reach the standard required for publication in ACP. Having re-assessed the reviewers' comments and conducted my own assessment, I have now concluded that a number of outstanding issues need to be addressed before reaching the required standard. Most importantly, it is insufficient to frame the complexity of potential aerosol effects on convection modulated via microphysics as "indirect effects" in the simplistic Albrecht framework. There exists a whole body of research and literature on these effects that cannot simply be ignored.*

*Please consider each of the items below in your response.*

*Kind regards,*

*Philip Stier*

*Co-Editor*

*Atmospheric Chemistry and Physics*

*Response P1:*

*"The first sentence of the abstract has been modified in order to be explicit: «Convective precipitation are known to be negatively affected by aerosol indirect effects through reduced precipitable water and convective instability, as stated in the previous literature.» "*

*The verdict on aerosol effects on convective precipitation through modulation of cloud microphysics is still open and very likely to be regime dependent. There exists no clear consensus in the literature so this statement will need to be either more general and inclusive or omitted.*

The first sentence of the abstract has been decomposed in the 2 following sentences:

Aerosols may impact precipitation in a complex way involving their direct and indirect effects. In a previous numerical study, the overall microphysical effect of aerosols was to weaken precipitation through reduced precipitable water and convective instability.

*Response P2:*

*"The following sentence has been added in the 'simulation experiment' section: « It is however important to keep in mind that the ranges that will be found in this study should be interpreted as an upper bound of aerosol indirect effects. »"*

*Given the significant structural uncertainties involved it is not appropriate to conclude on upper or lower bounds of the total effect from a single modelling study. Even if structural errors were negligible there exist a large number of parametric uncertainties that have not been sampled. The only dimension you sample is the perturbation strength so this need so be clear. However, given the extreme range of values chosen, these perturbations are likely to act as on/off switch for some processes, such as autoconversion. The implications should be discussed.*

The previous sentence were removed and replaced by the following text indicating the paramaterization uncertainties at extreme values. A discussion has also been added in the last paragraph of the conclusion (see next point).

On the flip side, extreme values of aerosol concentrations reach the bounds of permitted values in the microphysical scheme, suggesting that for these ranges of concentrations, the microphysical parameterizations may lead to more uncertainties.

*Response P4:*

*"This entire study is performed within WRF. That's OK, but I find little discussion of any possible limitations of that particular model. How broadly applicable do the authors think their results are? Are crucial elements still missing, even for WRF at such high resolution? (There is some discussion in the conclusions, but I would encourage expanding a bit on it.)*

*We expanded a bit the conclusion in order to reveal the main limitations of our model setting. A deeper discussion is made on the previous companion paper (Da Silva et al., 2018). "*

*Extensive work has been performed to understand uncertainties in cloud (resolving) modelling that should be acknowledged.*

The following paragraph has been added in the conclusion and acknowledges previous studies on understanding uncertainties in cloud modelling.

Due to the poor comprehension of aerosol indirect effects (Fan et al., 2016) and the known uncertainties associated with numerical modeling (Crétat et al., 2012 ; Diaconescu et al., 2007; Flaounas et al. 2011 ; Foley, 2010 ; Ramarohetra et al., 2015 ; Seth 1998), aerosol indirect effects have been found to be sensitive to the model configuration (Seifert and Beheng, 2006; Lebo and Seinfeld, 2011, Lebo et al., 2012; Fan et al., 2012; Morrison, 2012; ; Lebo and Morrison, 2014; Hill et al., 2015; Johnson et al., 2015; White et al., 2017; Heikenfeld et al., 2019). The results found with our specific model configuration are uncertain in this sense and need to be confirmed by further numerical and observational studies. It is also worth noting that these results were obtained using extremely low and extremely high aerosol concentrations. While this approach permits to effectively retrieve the precipitation response to a drastic change in aerosol concentrations, it is uncertain that the magnitude of the involved processes does not change under less extreme aerosol conditions, as stated in other studies (Rosenfeld et al., 2008; Dagan et al., 2015; Miltenberger et al., 2018). Despite these limitations, our configuration highlights the importance of the cloud albedo effect on convective precipitation, often underestimated in case studies simulations. It is suggested in this study that this effect might be higher than the convective invigoration, as predicted in Fan et al. (2013) using shorter simulations on a smaller domain. A more realistic estimation of the aerosol indirect effects on convective precipitation could be carried out with the use of online-coupled models in which aerosol concentrations are evaluated with precise emission and transport schemes (Tucella et al. 2019).

*Response P9:*

*"Precipitation extremes are supposed to wring out all of the moisture from an ascending parcel and are therefore expected to scale with the CC law. However many departures from the CC-scaling"*

*Statements as "supposed to" needs to be backed up by reference. "Wring out" does not correspond to a physical process. Language needs to refer to physical processes.*

The following correction has been made in the manuscript:

It has been suggested that precipitation extremes correspond to events where the whole column of moisture is precipitated and are therefore expected to scale with the CC law (Pall et al. 2007, Muller et al., 2013).

*Response P10:*
*"The CC scaling is less expected for mean precipitation which are more constrained by an energetic budget than extreme precipitation (Allen and Ingram, 2002; Held and Soden, 2006;*

*Muller et al., 2011; Muller, 2013). "*

*Please discuss the scales on which these constraints hold.*

A brief discussion has been added in the introduction.

While this constraint is theoretically more relevant at the global scale, it has also been observed by regional climate models (Raisanen et al., 2004). The study of Hardwick et al. (2010) suggests that this constraint also acts in a specific location over a long time record. Indeed, they have systematically found lower slopes for median precipitation than for extreme precipitation in their 4 selected in-situ measurement stations in Australia.

*Response P11 / Introduction:*
*You discuss "indirect effects" on precipitation in the framework of Abrecht etc. However, your results include the influence of convection for which a wide range of hypothesis exist regarding aerosol effects on precipitation due to potential interactions with the cloud dynamics, including the convective invigoration hypothesis. This needs to be reflected in the introduction and discussed later on as appropriate.*

Thank you for pointing out this point. Modifications has been added in the introduction, the results and the conclusion of the paper to discuss the potential effects that may hold in our simulations. However the added comments in the results and conclusion mostly consists in evocating hypotheses, a deeper analysis was considered beyond the scope of this paper.

Please see the differences between the previous version and new version of the manuscript at the end of the present document.

*Response P13:*
*You need to discuss which processes you may resolve in convection permitting simulation that you do not capture in parameterised convection. Presumably you now capture aerosol effects on convection? This then needs to be linked to the mechanism referred to in the last comment..*

The following sentence has been added in this part of the paper. Some reminders were also added in the result part when appropriate.

While the microphysical effects of aerosol on convective clouds were not taken into account due to the use of a convection scheme insensitive to aerosol concentrations, the whole set of indirect effects are represented in the HR simulations, including small scale and large scale processes.

*Response P13:*
*"at first vertical grid level" -> at lowest vertical grid level (the ordering in models is arbitrary)*

The text has been modified accordingly.

*Response P14:*
*Figure 2 and caption: it is unclear if results are all from same simulations/resolutions or not? How do they differ?*

We are not sure of the point but the figure was done using convective precipitation of the LR simulations (50 km of resolution), even inside subdomain boxes.

It has been specified in the caption.

*Response P16:*
*Figure 3: consider adding visual legend*

Visual legends have been added in figure 3 and 4 of the manuscript.

*Response P17:*
*"It confirms the weaker occurrence of clouds at high temperatures in our simulations, which results"*

*This sentence is confusing and needs to be clarified. What is "weaker occurrence"?*

This sentence has been replaced by the following sentence below. We hope that this sentence and the following added paragraph (see next comment), help in clarifying the text.

It shows a decrease of COD with temperatures in all of the simulations.

*Response P17*
*COD is not well defined here. Are we seeing a shift in cloud regimes or a change in cloud radiative properties within regimes? The shape is not obvious.*

Precisions have been added in the new version of the manuscript:

When averaging over hours with strictly positive COD, we found that the thickness of clouds is relatively constant over the temperature range (not shown), confirming that the decrease of COD with temperature is mostly due to a decrease of the occurrence of clouds with temperature. This tendency maximizes the indirect effects of aerosols at low temperatures and minimizes them at high temperatures.

*Response P17:*

*"We believe that the inhibition of convective precipitation is mainly due to the processes described in Da Silva et al. (2018), i.e. a stabilisation of the atmosphere and a reduction of precipitable water in the polluted simulations."*

*"Believe" is not appropriate in results section. Such statements need to be backed up by analysis / facts.*

The end of this paragraph has been rewritten with a starting discussion of precipitation efficiency, while the main analysis is delayed to the next section. This statement does not appear anymore in the new version of the manuscript.

Please see the differences between the previous version and new version of the manuscript at the end of the present document.

*Response P18:*
*"To analyse the reduction of convective precipitation at low temperatures we consider that precipitation can be approximately described by the following equation:"*

*The origin of this equation is not entirely obvious. Needs to be backed up by primary references.*

*"This description is mostly valid for convective precipitation which result from a parcel that raises from the surface."*

*Same here: such statements need to be referenced.*

These statements are founded on the work of Drobinski et al. (2016) and Da Silva (2018).

The references have been added in the manuscript.

*Response P18:*

*You discuss maximal vertical velocity in parameterised convection. Where does this come from? Not clear from the model description. Is this prognostic (unlikely)? How is it diagnosed?*

Maximal vertical velocities of the parameterised simulations were calculated using the square root of the CAPE, as precised in the method section of the manuscript.

*Response P21 :*
*"In this set of simulations with explicit convection, changes in aerosol concentrations may have an impact on precipitation efficiency through a change in autoconversion rate (second indirect effect). "*

*As discussed above, this is too simplistic for potential effects of aerosols on convective precipitation and ignores a wide body of research and literature.*

A new discussion is proposed and evocates some hypotheses of previous studies. Such simplistic statements were modified accordingly.

*Response P27 / conclusions:*
*"Figure 14 summarizes the various involved processes and their qualitative contribution (size of the arrows)."*

*It summarises the investigated processes but does not discuss other relevant processes (see above).*

The figure was slightly modified in order to account for a potential convective invigoration effect in our HR simulations, which was discussed in the result section.

*Response P29 / conclusions:*

*"These results should be interpreted as an upper bound of the aerosol climatological indirect effect on convective precipitation, since extremely and high aerosol concentrations were used in this study. A more realistic estimation of the aerosol indirect effect on convective precipitation could be carried out with the use of online-coupled models in which aerosol concentrations are evaluated with precise emission and transport schemes. Although taken into account in our simulations with explicit convection, our study suggests that the second aerosol indirect effect may not affect convective precipitation efficiency in a significant waycompared to the stabilisation effect. It is however likely that the second indirect effect plays a role in stabilising the atmosphere and hence in reducing convective precipitation, a result that remains to be established."*

*These statements are too simplistic and will need to be modified based on the points raised above.*

The last paragraph of the conclusion has been rewritten according to the previous remarks.

[revised manuscript text omitted]

---

## Author Response (AR3)

**Aerosol indirect effects on temperature-precipitation scaling**

*By N. Da Silva, S. Mailler and Ph. Drobinski*

Dear ACP Editor,

Thank you for your careful work which has led to substantial ameliorations of our manuscript. We have taken into account all of your new comments and modified the text accordingly. We hope that with these changes, our study will be retained for publication in ACP. In the text below, your comments are in italics, our answers in straight black fonts, and the text in blue describes (and generally reproduces) the changes that have been brought to the manuscript.

Kind Regards,

The Authors.

*Comments to the Author:*
*Dear Authors*

*Thank you for providing your revisions to the comments raised in the last iteration. While many of the comments have been addressed, there are still some outstanding issues that will need to be resolved before publication. Please address each of them in your response.*

*Best regards,*
*Philip Stier*

*Issues (page numbers from author response):*

*1) Response page 2: "On the flip side, extreme values of aerosol concentrations reach the bounds of permitted values in the microphysical scheme, suggesting that for these ranges of concentrations, the microphysical parameterizations may lead to more uncertainties."*

*This still (wrongly) suggests that the model could represent microphysics perfectly within the bounds of permitted values. Unfortunately, we do not yet have the sufficient confidence in mixed-phase cloud microphysics schemes to conclude this.*

This sentence was written to suggest that the uncertainties related to the parameterization formulation may be even larger at the bounds than within the bounds of permitted values, without excluding uncertainties within the bounds of permitted values. The sentence was reformulated into a (hopefully) more explicit assertion:

On the flip side, extreme values of aerosol concentrations reach the bounds of permitted values in the microphysical scheme, suggesting that for these ranges of concentrations, the uncertainties associated to the parameterizations of microphysical processes may be more pronounced.

*2) The discussion (and possibly understanding) of the energetic constrain appear incomplete. On page 9 it is suggested that "The study of Hardwick et al. (2010) suggests that this constraint also acts at a specific location over a long time record." This does not make sense as divergence of dry static energy can certainly balance any changes in latent heating on local scales (c.f. Muller and O'Gorman, 2011). Please make sure to reflect on this fact and update the manuscript accordingly.*

The transport term vanishes when integrating over the whole globe or over a sufficiently large area. It was suggested here (perhaps wrongly) that a similar reduction of the transport term could occur after a long temporal integration, by summing compensating convergence and divergence terms. Thank you for providing this reference which is indeed lacking in the discussion. According to their figure 1, transport can balance any change of latent heating on the local scales (especially over ocean), and thus can suppress the constraint in these regions. The discussion was therefore changed, taking into account the missing publication.

On the local scales, the energy budget includes a term accounting for the transport of dry static energy which may suppress the constraint in many regions, as shown by Muller and O'Gorman (2011). The study of Hardwick et al. (2010) suggests that Australia is part of the regions were the constraint still holds. Indeed, they have systematically found lower slopes for median precipitation than for extreme precipitation in their 4 selected in-situ measurement stations in Australia.

*3) Model uncertainties cannot be ignored. On page 10 you suggest that "A few studies fairly represented the overall indirect effects of aerosol by realising high resolution simulations over few months... . The study of Da Silva et al. (2018) (your own work... is one them". The wording fair vs unfair is subjective and it is clear that we are not yet aware of a perfect model - but know many uncertainties. Please update the wording accordingly based on physical arguments.*

We agree that the term 'fair' is not well used in this context. The use of an intermediate configuration may allow both effects to be represented but perhaps at the expense of increased uncertainties. In such intermediate configurations, the representation of the microphysical effects of aerosols on convective clouds may be more uncertain than for cloud resolving simulations, as well as few months of simulation may not be enough for fully represent the long-term radiative effect of aerosols. The text was modified in order to reflect the model uncertainties.

Few studies take into account both of the microphysical effects of aerosols on convection and the aerosol long-term radiative feedback by realising high resolution simulations over few months (Morrison and Grabowski, 2011; Fan et al. 2013), while not avoiding uncertainties related to their representation by RCM in these intermediate configurations.

*4) On page 17 you refer to "changes of vertical velocities between ... may be explained by the convective invigoration effect,". However, as acknowledged in the updated manuscript, there exist multiple potential invigoration mechanisms so there is not a single "invigoration effect".*

The term 'convective invigoration effect' indeed does not refer to a single process, it was therefore removed from this sentence:

For extreme events, which mostly consist of convective events, the discrepancy between the strong changes of CAPE and the weaker changes of vertical velocities between the HR MAX and the HR MIN simulation may be explained by enhanced release of latent heat at the freezing level caused by increased vertical mass transport of water droplets in polluted conditions, as suggested by previous studies (Khain et al., 2004, 2005; Rosenfeld et al., 2008; Lebo and Seinfeld, 2011; Fan et al., 2013).

*5) Conclusions. You conclude on page 30 that "Despite these limitations, our configuration highlights the importance of the cloud albedo effect on convective precipitation". Presumably you mean due the surface dimming and subsequent reduction of surface fluxes? This should be clear and the physical mechanism explained. Would this then not hold for precipitation in general? The current wording seems to suggest a direct link between cloud albedo and convective precipitation, which is confusing.*

The sentence was rephrased in order to suggest that the link between aerosol radiative feedback and convective precipitation is not direct. Since the causal chain was already described earlier in the conclusion, we suggest to evoke it through the general term 'changes of the thermodynamic profile of the atmosphere'. Whether or not this effect is relatively important in changing non-convective precipitation with respect to other aerosol indirect effects remains a question. Within the same model configuration, the explicit precipitation of the low resolution simulation (50 km of horizontal grid spacing) were shown to slightly increase with increased aerosol concentrations (Da Silva et al., 2018), suggesting a less important reduction of non-convective precipitation through this path.

Despite these limitations, our configuration highlights the importance of the background aerosol cloud radiative feedback and its repercussions on convective precipitation through changes of the thermodynamic profile of the atmosphere, often underestimated in case studies simulations.

*6) The revised section contain a number of grammatical errors that should be fixed before publication.*

These grammatical errors were corrected in the new version of the manuscript.

[revised manuscript text omitted]